# Stable machine-learning parameterization of subgrid processes for climate modeling at a range of resolutions

Janni Yuval [1]✉ & Paul A. O'Gorman[1]

Global climate models represent small-scale processes such as convection using subgrid models known as parameterizations, and these parameterizations contribute substantially to uncertainty in climate projections. Machine learning of new parameterizations from high-resolution model output is a promising approach, but such parameterizations have been prone to issues of instability and climate drift, and their performance for different grid spacings has not yet been investigated. Here we use a random forest to learn a parameterization from coarse-grained output of a three-dimensional high-resolution idealized atmospheric model. The parameterization leads to stable simulations at coarse resolution that replicate the climate of the high-resolution simulation. Retraining for different coarse-graining factors shows the parameterization performs best at smaller horizontal grid spacings. Our results yield insights into parameterization performance across length scales, and they also demonstrate the potential for learning parameterizations from global high-resolution simulations that are now emerging.

[1] Massachusetts Institute of Technology, Cambridge, MA 02139, USA. ✉email: janniy@mit.edu

Coupled atmosphere–ocean simulations of climate typically resolve atmospheric processes on horizontal length scales of order 50–100 km. Smaller-scale processes, such as convection, are represented by subgrid parameterization schemes that typically rely on heuristic arguments. Parameterizations are a main cause for the large uncertainty in temperature, precipitation and wind projections[1–6]. Although increases in computational resources have now made it possible to run simulations of the atmosphere that resolve deep convection on global domains for periods of a month or more[7,8], such simulations cannot be run for the much longer timescales over which the climate system responds to radiative forcing[9], and the computational cost to explicitly resolve important low cloud feedbacks will remain out of reach for the foreseeable future[6]. Therefore, novel and computationally efficient approaches to subgrid parameterization development are urgently needed and are at the forefront of climate research.

Machine learning (ML) of subgrid parameterizations provide one possible route forward given the availability of high-resolution model output for use as training datasets[10–16]. While high-resolution simulations still suffer from biases, to the extent that they resolve atmospheric convection, a parameterization learned from these simulations has the potential to outperform conventional parameterizations for important statistics such as precipitation extremes. Training on both the control climate and a warm climate is needed to simulate a warming climate using an ML parameterization[12,13], and this is feasible because only a relatively short run of a high-resolution model is needed for training data in the warmer climate.

ML parameterization could also have advantages for grid spacings that are smaller than in current global climate models (GCMs) but not yet convection resolving. At these gray-zone grid spacings, assumptions traditionally used in conventional parameterizations, such as convective quasi-equilibrium[17], may need to be modified or replaced such that the parameterization is scale aware[18,19]. Without such modifications, it may be better to turn off some conventional parameterizations of deep convection for a range of grid spacings that are too close to the convective scale[20,21]. Since ML parameterizations can be systematically trained at different grid spacings without the need to change physical closure assumptions, an ML approach to parameterization has the potential to perform well across a range of grid spacings and to provide insights into the scale dependence of the parameterization problem.

Recently a deep artificial neural network (NN) was successfully used to emulate the embedded two-dimensional cloud-system resolving model in a superparameterized climate model in an aquaplanet configuration[11,12], although some choices of NN architecture could lead to instability and blow ups in the simulations[22]. An NN parameterization has also been recently learned from the coarse-grained (spatially averaged to a coarser grid) output of a fully three-dimensional model, with issues of stability dealt with by including multiple time steps in the training cost function and by excluding upper-tropospheric levels from the input features[14,15]. This NN parameterization could be used for short-term forecasts, but it suffered from climate drift on longer times scales and could not be used for studies of climate. Thus, an ML parameterization has not yet been successfully learned from a three-dimensional high-resolution atmospheric model for use in studies of climate.

One approach that may help the robustness and stability of an ML parameterization is to ensure that it respect physical constraints such as energy conservation[23]. Using a random forest (RF)[24,25] to learn a parameterization has the advantage that the resulting parameterization automatically respects energy conservation (to the extent energy is linear in the predicted quantities) and non-negative surface precipitation[13]. An RF is an ensemble of decision trees, and the predictions of the RF are an average of the predictions of the decision trees[24,25]. Physical constraints are respected by an RF parameterization because the predictions of the RF are averages over subsets of the training dataset. The property that the RF predictions cannot go outside the convex hull of the training data may also help ensure that an RF parameterization is robust when implemented in a GCM. When an RF was used to emulate a conventional convective parameterization, it was found to lead to stable and accurate simulations of important climate statistics in tests with an idealized GCM[13]. Thus RFs are promising for use in learning parameterizations of atmospheric processes, but they have not yet been used to learn subgrid moist processes from a high-resolution atmospheric model.

In this study we learn an RF parameterization from coarse-grained output of a high-resolution three-dimensional model of a quasi-global atmosphere, and we show that the parameterization can be used at coarse resolution to reproduce the climate of the high-resolution simulation. By learning different RF parameterizations for a range of coarse-graining factors, we assess the performance of the RF parameterization as the grid spacing in the coarse model is varied, and this helps addresses the important question of over what range of coarse-graining factors an ML parameterization of convection can be successful.

## Results

**Learning from high-resolution model output.** The model used is the System for Atmospheric Modeling (SAM)[26], and the domain is an equatorial beta plane of zonal width 6912 km and meridional extent 17,280 km in an aquaplanet configuration. The distribution of sea surface temperature (SST) is specified to be zonally and hemispherically symmetric and reaches a maximum at the equator (the qobs SST distribution[27]). To reduce computational expense, we use hypohydrostatic rescaling (with a scaling factor of 4) which effectively increases the horizontal length scale of convection and allows us to use a coarser horizontal grid spacing of 12 km than would be normally used in a cloud-system resolving simulation, while not affecting the large-scale dynamics[28–31]. Further details of the model configuration are given in the methods section.

The high-resolution simulation (hi-res) exhibits organization on a wide range of length scales from the convective to the planetary scale (Fig. 1a). The largest-scale organization consists of two intertropical convergence zones (ITCZs) and an extratropical storm track in the mid-latitudes of each hemisphere. The configuration used here in which the SST distribution is fixed and symmetric about the equator is a challenging test of our RF parameterization since the resulting circulation is known to be very sensitive to subgrid parameterizations, and coarse-resolution GCMs in this configuration give a range of tropical circulations from a strong single ITCZ to a double ITCZ[32]. We find there is a double ITCZ at high resolution for our model configuration, and this is likely dependent on the exact SST distribution used and the geometry of the domain. When the model is run with a horizontal grid spacing of 96 km and thus eight times coarser horizontal resolution (x8), the double ITCZ switches to a much stronger single ITCZ (Fig. 1b) and the distribution of mean precipitation is strongly altered throughout the tropics (Fig. 2a). Extreme precipitation, which is important for impacts on society and ecosystems, is evaluated here as the 99.9th percentile of 3-hourly precipitation; it is sensitive at all latitudes to changing from high to coarse resolution (Fig. 2b). In this study, we do not compare the results of the hi-res simulation to a coarse-resolution simulation with conventional convective and boundary-layer

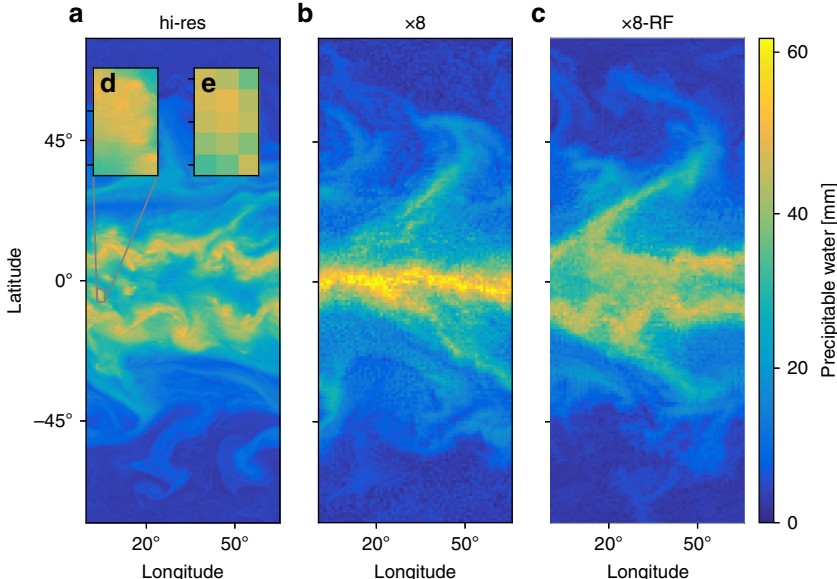

**Fig. 1 Snapshots of column-integrated precipitable water taken from the statistical equilibrium of simulations. a** High-resolution simulation (hi-res),
**b** coarse-resolution simulation (x8), and **c** coarse-resolution simulation with random forest (RF) parameterization (x8-RF). Insets in **a** show **d** a zoomed-in
region and **e** the same region but coarse-grained by a factor of 8 to the same grid spacing as in **b**. The colorbar is saturated in parts of panel **b**.

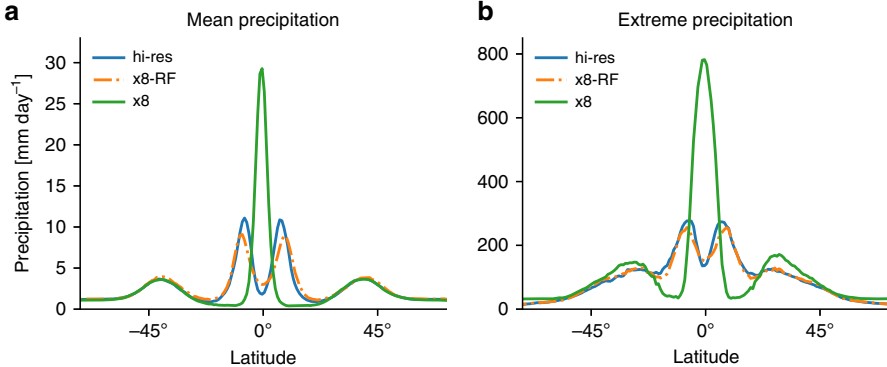

**Fig. 2 Mean and extreme precipitation as a function of latitude. a** Zonal- and time-mean precipitation and **b** 99.9th percentile of 3-hourly precipitation,
for the high-resolution simulation (hi-res; blue), and the coarse resolution simulation with the random forest (RF) parameterization (x8-RF; orange
dashed–dotted) and without the RF parameterization (x8; green). For hi-res, the precipitation is coarse-grained to the grid spacing of x8 prior to calculating
the 99.9th percentile to give a fair comparison[41].

parameterizations both because SAM is not equipped with such
parameterizations and because the results in the tropics would be
highly dependent on the specific choice of parameterizations for
both mean precipitation[32] and extreme precipitation[4].

The RF parameterization predicts the effect of unresolved
subgrid processes that act in the vertical, including vertical
advection, cloud and precipitation microphysics, vertical turbu-
lent diffusion, surface fluxes and radiative heating, on the resolved
thermodynamic and moisture prognostic variables at each
gridbox and time step. The prognostic variables that are explicitly
affected by the RF-parameterization are the liquid/ice water moist
static energy ($h_L$), total non-precipitating water mixing ratio ($q_T$),
and precipitating water mixing ratio ($q_P$). Subgrid momentum
fluxes are not predicted, but this is not expected to strongly affect
the results since we do not have topography that could generate
strong gravity wave drag and since tropical convection occurs in
regions of relatively weak shear in our simulations. We assume
that the subgrid contributions depend only on the vertical column
of the grid point at the current time step, and we predict all

outputs in the vertical column together, and therefore the
parameterization is column based and local in time and in the
horizontal. We chose to use two RFs so that we can separately
predict processes (turbulent diffusion and surface fluxes) that
depend on horizontal winds and are primarily active at lower
levels of the atmosphere.

The first RF, referred to as RF-tend, predicts the vertical
profiles at all 48 model levels of the combined tendencies due
to subgrid vertical advection, subgrid cloud microphysics,
subgrid sedimentation and falling of precipitation, and total
radiative heating. Hence the outputs of RF-tend are $Y_{RF-tend} =
(h_L^{subg-tend}, q_T^{subg-tend}, q_P^{subg-tend})$ where subg-tend refers to the
subgrid tendency, giving $48 \times 3 = 144$ outputs. Radiative heating
is treated as entirely subgrid, whereas the other processes have a
resolved representation on the coarse model grid and a subgrid
component represented by the RF parameterization. We do not
use the RF-parameterization to predict radiative heating for levels
above 11.8 km because it does not predict the radiative heating

well for those levels, possibly because of insufficient coupling between the stratosphere and troposphere. Subgrid tendencies for vertical advection and microphysics are calculated as the horizontal coarse-graining of the tendencies at high resolution minus the tendencies calculated from the model physics and dynamics using the coarse-grained prognostic variables as inputs (see methods). The features (inputs) for RF-tend ($X_{RF-tend}$) are chosen to be the vertical profiles (discretized on model levels) of the resolved temperature ($T$), $q_T$, $q_P$, and the distance from the equator ($|y|$). Hence $X_{RF-tend} = (T, q_T, q_P, |y|)$, giving $48 \times 3 + 1 = 145$ features. Distance from the equator serves as a proxy for the SST, surface albedo and solar insolation, as these are only a function of this distance in the simulations considered here. For a different simulation setup that is not hemispherically symmetric, we would include these physical quantities as separate features instead of distance from the equator.

The second RF, referred to as RF-diff, predicts the coarse-grained turbulent diffusivity ($\overline{D}$) for thermodynamic and moisture variables and the subgrid correction to the surface fluxes. We predict $\overline{D}$ rather than the turbulent diffusive tendencies so as to ensure that the turbulent fluxes remain downgradient. For computational efficiency we only predict $\overline{D}$ in the lower troposphere (the 15 model levels below 5.7 km) because it decreases in magnitude with height (Supplementary Fig. 1d). Hence the outputs of RF-diff are $Y_{RF-diff} = (\overline{D}, h_L^{surf-flux}, q_T^{surf-flux})$ where surf-flux refers to a subgrid surface flux, giving $15 + 1 + 1 = 17$ outputs. The features of RF-diff are chosen to be the lower tropospheric vertical profiles of $T$, $q_T$, zonal wind ($u$), meridional wind ($v$), surface wind speed ($wind_{surf}$), and distance from the equator, so that $X_{RF-diff} = (T, q_T, u, v, wind_{surf}, |y|)$, giving $4 \times 15 + 1 + 1 = 62$ features. Since the meridional velocity is statistically anti-symmetric with respect to reflection about the equator, the meridional wind in the southern hemisphere is multiplied by $-1$ when it is taken as a feature for RF-diff to help ensure that RF-diff is not learning non-physical relationships between inputs and outputs that could artificially improve our results. We include the wind variables as features for RF-diff because they improve the prediction of the diffusivity and subgrid surface fluxes. Adding wind features to RF-tend does not improve the accuracy of the predicted tendencies.

The methods section gives further details about the RFs. In Supplementary Note 1 we demonstrate that the RF parameterization respects the physical constraints of energy conservation (Supplementary Fig. 2) and non-negative surface precipitation (Supplementary Fig. 3).

**Simulation with RF parameterization**. A simulation with the RF parameterization at 96 km grid spacing (x8-RF) was run using an initial condition taken from the statistical equilibrium of the x8 simulation with no RF parameterization. The x8-RF simulation transitions to a new statistical equilibrium with a double ITCZ similar to that in the high-resolution simulation (Fig. 1c) and it runs stably over long timescales (we have run it for a 1000 days). At statistical equilibrium, the distribution of mean precipitation is close to that of the high-resolution simulation (Fig. 2a), and the distribution of extreme precipitation is remarkably well captured (Fig. 2b). Other measures such as eddy kinetic energy, mean zonal wind, mean meridional wind and mean $q_T$ are also correctly captured by x8-RF (Supplementary Table 1). Overall, these results show that using the RF subgrid parameterization brings the climate of the coarse-resolution simulation into good agreement with the climate of the high-resolution simulation.

The x8-RF simulation requires roughly 30 times less processor time than the high-resolution simulation (for x16-RF the speed up is by roughly a factor of 120). Further increases in speed could be obtained by increasing the time step but this is limited in part by the fall speed of precipitation. In Supplementary Note 2, we present an alternative RF parameterization in which $q_P$ is no longer treated as a prognostic variable and which could be used to achieve even faster simulations at coarse resolution in future work. This alternative parameterization has comparable performance to our default parameterization (Supplementary Fig. 4) but it requires certain outputs to be set to zero (above 11.8 km) to avoid a deleterious feedback possibly related to an issue of causality when $q_P$ is not evolved forward in time[15], and it is less accurate for extreme precipitation in mid-latitudes.

**Performance for different horizontal grid spacings**. The fact that the RF parameterization is learned from a fully three-dimensional simulation with a wide range of length scales allows us to explore the question of whether there is a particular range of grid spacings for which an ML parameterization could be most successful. With increasing grid spacing, coarse-graining involves more averaging over different cloud elements which should make the subgrid tendencies more predictable, but the parameterization is then also responsible for more of the dynamics and physics.

We train RF parameterizations for a range of coarse-graining factors from x4 to x32 and use them in simulations with corresponding grid spacings. We first describe the performance of the RFs on offline tests (i.e., when the RFs are not implemented in SAM) based on data withheld in training. The offline performance as measured by the coefficient of determination ($R^2$) improves substantially as the grid spacing increases (Fig. 3a, compare Fig. 3c, e), consistent with the idea of more predictable subgrid tendencies with more averaging over larger grid boxes. Improved offline performance with increasing grid spacing is shown to hold for all of the predicted outputs in Supplementary Table 2.

However, online performance (i.e., the ability of the coarse-resolution simulations with the RF parameterization to correctly capture the climate of hi-res) varies very differently with grid spacing as compared to offline performance. Online performance increases monotonically with decreasing grid spacing (Fig. 3b and compare Fig. 3d, f), and the best performance is found for x4-RF, indicating that the RF parameterization can work well with a relatively small gap between the grid spacings of the coarse-resolution model and the high-resolution model from which it was learned.

One might think the decrease in online performance at larger grid spacings is due to more of the subgrid dynamics and physics becoming subgrid and thus the absolute errors in the predicted subgrid tendencies becoming larger even if $R^2$ increases, but the root mean square error (RMSE) in offline tests actually decreases as the grid spacing increases (Fig. 3a and Supplementary Table 3). To understand the discrepancy between variations in offline and online performance, it is helpful to think of the variables that the RFs predict as having two components—a predictable component and a stochastic component. For smaller grid spacing, the stochastic component is large (compare the same snapshot for different coarse-graining factors in Fig. 4a, c), and the prediction task becomes more difficult (compare Fig. 4d, f). Therefore, the relatively low offline $R^2$ at smaller grid spacing does not necessarily imply that the RF does not predict the predictable component accurately. To demonstrate this point we make a comparison between offline performance of x4-RF and x32-RF, but we first coarse grain the subgrid tendencies calculated and predicted at x4 to the x32 grid, and we refer to the results of this

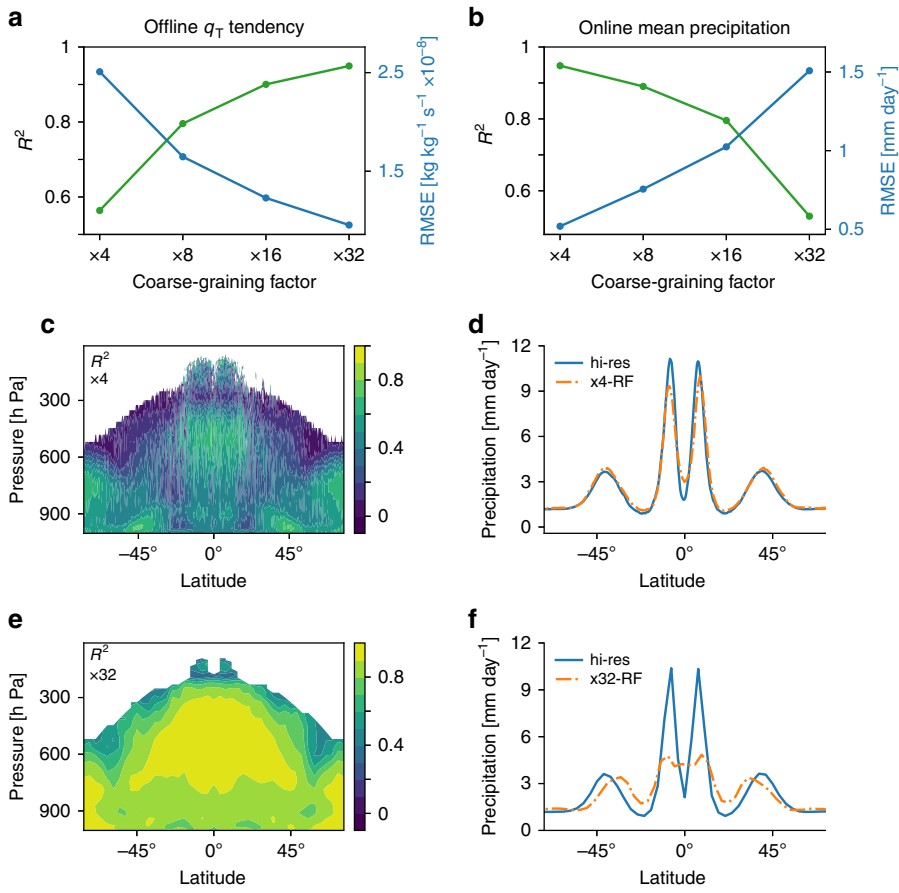

**Fig. 3 Performance of random forest parameterization versus grid spacing.** Panels **a**, **c** and **e** show offline performance as measured on test data for the random forest (RF) predicted tendency of $q_T$: **a** $R^2$ (green) and root mean square error (blue) versus grid spacing, and **c**, **e** $R^2$ versus pressure and latitude for **c** x4 and **e** x32. In **c**, **e**, $R^2$ is only shown where the variance is at least 0.1% of the mean variance over all latitudes and levels. Panels **b**, **d** and **f** show online performance: **b** R2 (green) and root mean square error (blue) versus grid spacing for mean precipitation versus latitude, and **d**, **f** mean precipitation versus latitude for hi-res (blue) compared to **d** x4-RF (orange) and **f** x32-RF (orange).

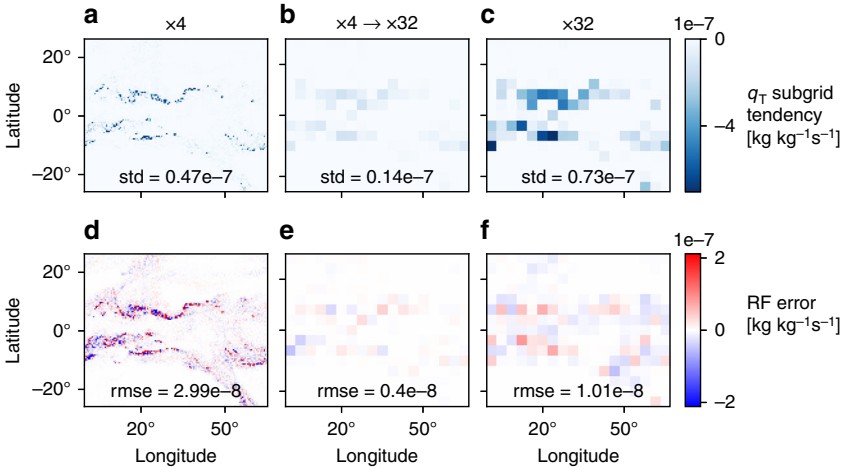

**Fig. 4 Offline comparison of random forest parameterizations at a common grid spacing.** Snapshot of the subgrid tendency of $q_T$ at 5.7 km in kg kg$^{-1}$ s$^{-1}$ showing **a**–**c** the true tendency and **d**–**f** the error in the prediction from RF-tend. Results are shown for **a**, **d** subgrid tendencies calculated and predicted at x4, **b**, **e** coarse graining of the subgrid tendencies calculated and predicted at x4 to x32 grid spacing (x4 → x32), and **c**, **f** subgrid tendencies calculated and predicted at x32. Inset text gives for the snapshot shown **a**–**c** the standard deviation of the true tendencies and **d**–**f** the root mean square error.

procedure as x4 → x32 (Fig. 4b and methods). The RMSE for x4 → x32 is substantially smaller than for x4 because the stochastic component averages out with coarse-graining (compare Fig. 4a, b). Importantly the RMSE for x4 → x32 is also substantially smaller than for x32 (compare Fig. 4e, f). Therefore, x4-RF has smaller offline errors compared to x32-RF when these parametrizations are compared in an apples-to-apples comparison at the same length scale, and this is consistent with the better

online performance of x4-RF than x32-RF. Similar results are found for other vertical levels and other outputs of the RF (Supplementary Fig. 5 and Supplementary Table 4). We note that for some outputs $R^2$ is still higher for x32 than x4 → x32 (Supplementary Table 4), and thus it seems it is more appropriate in this case to compare the absolute rather than relative errors of the parameterization.

For extreme precipitation, the improvement in online performance with decreasing grid spacing is weaker than for mean precipitation, and there is no improvement in extreme precipitation performance when decreasing the grid spacing from x8 to x4 (Supplementary Fig. 6) indicating the possibility of a slight gray zone for this statistic. Nonetheless, we conclude there is a clear overall improvement in online performance of the RF parameterization as grid spacing decreases, which suggests that ML parameterizations could be useful for grid spacings that are quite close to that of the high-resolution model from which they are learned.

**Robustness of the RF parameterization**. We performed tests to check the robustness of the RF parameterization, and in particular to confirm that its skill is not based on learning that particular circulations (such as ascent in an ITCZ) and associated clouds occur at particular latitudes. As a first test, we re-trained the RF parameterization without using the distance from equator as a feature since much of the subgrid dynamics and physics (e.g., vertical advection, cloud, and precipitation microphysics, and longwave cooling) represented by the RF parameterization should be largely predictable from features other than distance to the equator (which is a proxy for surface albedo, insolation, and SST). We find that the offline results are similar regardless of whether distance from the equator is used as a feature in the RFs (Supplementary Tables 2, 3).

As a second test, we trained new versions of the RF parameterization in which latitudes bands of width 10° were excluded during training in both hemispheres, and thus the RF parameterization must generalize across latitudes when it is used in simulations. Based on offline tests we find that x8-RF can generalize remarkably well when tropical latitude bands containing the ITCZs are excluded (Fig. 5a), and there is only a slight decrease in performance for excluded latitude bands at high latitudes (Fig. 5c). Excluding latitude bands in mid-latitudes leads to a marked deterioration in performance (Fig. 5b), likely due to the small overlap between the features in the center of the excluded latitude bands and the training data outside the latitude bands. The lack of feature overlap in the midlatitude case is due to strong meridional gradients in temperature and mixing ratios, and lapse rates and relative humidity could be used as alternative features to avoid this overlap problem in future work. The resulting climates in coarse-resolution simulations with these RF parameterizations are remarkably similar (with a slight exception for precipitation in the midlatitude case) to the climate obtained using x8-RF trained on all the latitudes (Fig. 5d–f and Supplementary Fig. 7).

The results of these tests suggest that the success of the RF parameterization is not based on learning that particular circulation features occur at particular latitudes (for example, the RF parameterization is successful even when its training excludes the ITCZ regions), but rather it is learning robust physical relationships between features and outputs.

## Discussion
The results presented here provide a step forward by demonstrating the viability of stable and accurate parameterizations of subgrid physics and dynamics learned from a high-resolution three-dimensional simulation of the atmosphere. The results also

give insights into how well an ML parameterization can perform as a function of grid spacing. Online performance improves with decreasing grid spacing of the coarse-resolution model. This is in contrast to the experience that some conventional parameterizations are best turned off for a range of length scales that are too close to the convective scale[20,21], and the difference may arise because conventional parameterizations rely on physical assumptions that are not uniformly valid across length scales, although this can be mitigated by trying to make such parameterizations scale aware[18,19,33]. Care is needed in comparing offline performance across grid spacings, and we find that it is useful to compare offline error statistics at a consistent reference length scale. Further work using a model without hypohydrostatic scaling would be helpful to further investigate the behavior of ML parameterizations at different grid spacings.

The approach to ML parameterization for the atmosphere in this study is different in important aspects to previous studies. First, the predicted tendencies are calculated accurately for the instantaneous atmospheric state rather than approximating them based on differences over 3-hour periods[14,15]. Second, the subgrid corrections are calculated independently for each physical process rather than for all processes together as in previous studies[12,14,15] which allows for an ML parameterization structure that is motivated by physics and the calculation of the precipitation rate from the predicted tendencies. Third, we use an RF to learn from a high-resolution model whereas NNs have been used in previous studies that learned from a high-resolution model[10,12,14,15] and RFs were used only to emulate conventional parameterizations[13,34]. Parameterizations based on an RF have advantages in that their predictions automatically satisfy physical properties in the training data (without being imposed explicitly[23]) and they make conservative predictions for samples outside of the training data which may help with the robustness of their online performance. On the other hand, NNs require less memory and may have better offline performance. To further compare RF and NN parameterizations, future work should evaluate their online and offline performance using the same training data and atmospheric model.

Future research on ML parameterization for the atmosphere must address technical challenges such as how best to train over land regions with topography and how best to deal with the need for a separate parameterization of radiative heating in the stratosphere. However, future research should also continue to seek insights into the nature of the parameterization problem, such as how performance varies across length scales or whether parameterizations should be nonlocal in time and space, which may also inform the further development of conventional parameterizations.

## Methods
**Model**. The model used in this study is SAM version 6.3[26], which is a relatively efficient model that integrates the anelastic equations of motion in Cartesian coordinates. The bulk microphysics scheme is single moment with precipitating water consisting of rain, snow and graupel, and non-precipitating water consisting of water vapor, cloud water, and cloud ice. Cloud ice experience sedimentation, and we include the surface sedimentation flux (which is small) in all reported surface precipitation statistics. The subgrid-scale turbulent closure is a Smagorinsky-type scheme. The radiation scheme is based on parameterizations from the National Center for Atmospheric Research (NCAR) Community Climate Model (CCM) version 3.5[35].

The equations for the prognostic thermodynamic and moisture variables in SAM are important for our study and may be written as[26]

$$\frac{\partial h_{\mathrm{L}}}{\partial t} = -\frac{1}{\rho_0}\frac{\partial}{\partial x_i}(\rho_0 u_i h_{\mathrm{L}} + F_{h_{\mathrm{L}}i}) - \frac{1}{\rho_0}\frac{\partial}{\partial z}(L_{\mathrm{p}}P_{\mathrm{tot}} + L_{\mathrm{n}}S) + \left(\frac{\partial h_{\mathrm{L}}}{\partial t}\right)_{\mathrm{rad}}, \quad (1)$$

$$\frac{\partial q_{\mathrm{T}}}{\partial t} = -\frac{1}{\rho_0}\frac{\partial}{\partial x_i}(\rho_0 u_i q_{\mathrm{T}} + F_{q_{\mathrm{T}}i}) + \frac{1}{\rho_0}\frac{\partial}{\partial z}(S) - \left(\frac{\partial q_{\mathrm{p}}}{\partial t}\right)_{\mathrm{mic}}, \quad (2)$$

$$\frac{\partial q_{\mathrm{p}}}{\partial t} = -\frac{1}{\rho_0}\frac{\partial}{\partial x_i}(\rho_0 u_i q_{\mathrm{p}} + F_{q_{\mathrm{p}}i}) + \frac{1}{\rho_0}\frac{\partial}{\partial z}(P_{\mathrm{tot}}) + \left(\frac{\partial q_{\mathrm{p}}}{\partial t}\right)_{\mathrm{mic}}, \quad (3)$$

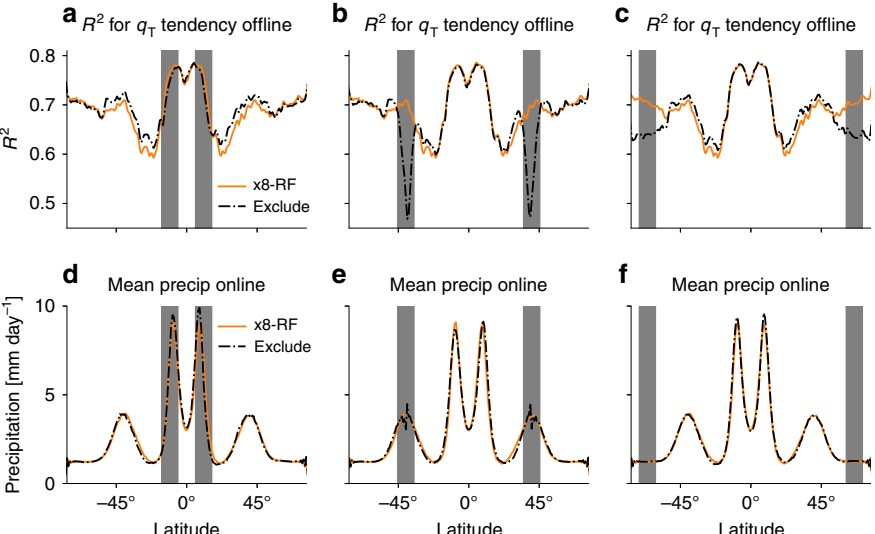

**Fig. 5 Performance of random forest parameterization when excluding latitude bands during the training process. a–c** Offline performance at x8 as measured by R$^2$ of the tendency of $q_T$ (including all longitudes and levels) versus latitude, and **d–f** online performance as shown by the zonal- and time-mean precipitation for the x8-RF simulation. Results are shown in black dash-dotted for cases in which the training process excludes in both hemispheres the latitude bands **a**, **d** 5.1°–15.5°, **b**, **e** 34.5°–44.9°, and **c**, **f** 60.5°–70.8°. Gray bars indicate latitude bands that where excluded during the training. For comparison the results for x8-RF without any latitudes excluded in training are plotted in orange.

where $h_L = c_p T + gz - L_c(q_c + q_r) - L_s(q_i + q_s + q_g)$ is the liquid/ice water static energy; $\rho_0(z)$ is the reference density profile; $q_T$ is the non-precipitating water mixing ratio which is the sum of the mixing ratios of water vapor ($q_v$), cloud water ($q_c$) and cloud ice ($q_i$); $q_p$ is the total precipitating water mixing ratio which is the sum of the mixing ratios of rain ($q_r$), snow ($q_s$) and graupel ($q_g$); $F_{Ai}$ is the diffusive flux of variable $A$; $u_i = (u, v, w)$ is the three-dimensional wind; $P_{tot}$ is the total precipitation mass flux (defined positive downwards); $S$ is the total sedimentation mass flux (defined positive downwards); the subscript rad denotes the tendency due to radiative heating; the subscript mic represents the microphysical tendency due to autoconversion, aggregation, collection, and evaporation and sublimation of precipitation; $L_c$, $L_f$ and $L_s$ are the latent heat of condensation, fusion and sublimation, respectively; $L_p = L_c + L_f(1 - \omega_p)$ is the effective latent heat associated with precipitation, and $\omega_p$ is the partition function for precipitation which determines its partitioning between liquid and ice phases; $L_n = L_c + L_f(1 - \omega_n)$ is the effective latent heat associated with non-precipitating condensate, and $\omega_n$ is the partition function for non-precipitating condensate which determines its partitioning between liquid and ice phases. Note that we do not introduce any prescribed large-scale tendencies in our simulations.

**Simulations.** All simulations are run on the same quasi-global domain with an equivalent latitude range from −78.5° to 78.5° and longitudinal extent of 62.2° at the equator. There are 48 vertical levels with spacing that increases from 85 m at the surface to 1650 m in the stratosphere, and the top level is at 28,695 m. The default time step is 24 s, and this is adaptively reduced as necessary to prevent violations of the CFL condition. The insolation is set at perpetual equinox without a diurnal cycle. The simulations are run with a zonally symmetric qobs[27] SST distribution which varies between 300.15 K at the equator and 273.15 K at the poleward boundaries. Surface albedo is a function of latitude, and there is no sea ice in the model. Simulations with a diurnal cycle and different SST distributions should be investigated in future work.

Hypohydrostatic rescaling of the vertical momentum equation with a rescaling factor of 4 increases the horizontal length scale of convection while leaving the large-scale dynamics unaffected and still retaining a very large range of length scales in the hi-res simulation[28–31,36]. A similar configuration of SAM with hypohydrostatic rescaling (though not at equinox) was recently used to investigate tropical cyclogenesis in warm climates[31]. Furthermore, SAM was also used in previous studies that developed ML parameterizations[12,14,15].

The hi-res simulation has 12 km grid spacing (recalling that hypohydrostatic rescaling is used) and was spun up for 100 days. It was then run for 500 days with three-dimensional snapshots of the prognostic variables, radiative heating, and turbulent diffusivity saved every 3 h. Results for the hi-res simulation are averaged over 500 days. Coarse-resolution simulations were run for 600 days, with the first 100 days of each simulation treated as spinup, and results averaged over the last 500 days. Simulations with the RF parameterization start with initial conditions taken from simulations without the RF parameterization (at the same resolution). The transition in the simulations with the RF parameterization from a single ITCZ in the initial condition to a double ITCZ sometimes occurs in two distinct steps,

but a spinup period of 100 days was found to be sufficient for this transition to occur.

The version of SAM that was used for the hi-res simulation had some minor discretization errors, the most important of which were in the Coriolis parameter in the meridional momentum equation and in the momentum surface fluxes. Effectively the Coriolis parameter is shifted by a distance of half a gridbox (6 km) to the south, and the surface winds used for calculating the surface momentum fluxes are also shifted by a distance of half a gridbox (but each wind component in a different direction). We corrected these errors when running the coarse-resolution simulations since discretization errors become larger in magnitude with coarser grid spacing. To avoid wasteful rerunning of the expensive hi-res simulation, in all coarse-resolution simulations we also shifted the Coriolis parameter in the meridional momentum equation by 6 km (half of the hi-res gridbox size) and shifted the surface winds by 6 km when calculating the surface momentum fluxes such that the coarse-resolution simulations are completely consistent with the hi-res simulation.

**Coarse graining and calculation of subgrid terms.** For each 3-hourly snapshot from the hi-res simulation, we coarse grain the prognostic variables $(u, v, w, h_L, q_T, q_p)$, the tendencies of $h_L$, $q_T$, and $q_p$ (eqs. (1)–(3)), the surface fluxes and the turbulent diffusivity. Coarse-graining is performed by horizontal averaging onto a coarser grid as follows:

$$\overline{A}(i, j, k) = \frac{1}{N^2} \sum_{l=N(i-1)+1}^{l=Ni} \sum_{m=N(j-1)+1}^{m=Nj} A(l, m, k), \qquad (4)$$

where $A$ is the high-resolution variable, $\overline{A}$ is the coarse-grained variable, $N$ is the coarse graining factor, $k$ is the index of the vertical level, and $i, j$ ($l, m$) are the discrete indices of the longitudinal and latitudinal coordinates at coarse resolution (high resolution).

Different coarse-graining factors were used to study how well the ML parameterization performs at different resolutions. The horizontal grid spacings that were used were 48 km (×4), 96 km (×8), 192 km (×16), and 384 km (×32). The hi-res simulation has a grid size of 576×1440, and coarse graining it by factors of 4, 8 and 16 results in grid sizes of 144 × 360, 72 × 180 and 36 × 90, respectively. These grids can be simulated in SAM. Unfortunately, coarse-graining the hi-res simulation by a factor of 32 results in a grid (18 × 45) which cannot run in SAM. Instead, the number of grid points in the latitudinal direction in these simulations was increased to 48 points (18×48 grid size), leading to a slightly larger domain, and the presented results were interpolated to the coarse-grained high-resolution grid (with 45 points in the latitudinal direction).

We define the resolved tendency as the tendency calculated using the dynamics and physics of model with the coarse-grained prognostic variables as inputs. The tendencies due to unresolved (subgrid) physical processes were calculated as the difference between the coarse-grained tendency and the resolved tendency. The

subgrid tendency for a given process is then written as

$$\left(\frac{\partial \overline{B}}{\partial t}\right)^{\text{subgrid}} = \frac{\partial \overline{B}}{\partial t}(h_{\text{L}}, q_{\text{T}}, q_{\text{p}}, u, v, w) - \frac{\partial B}{\partial t}(\overline{h}_{\text{L}}, \overline{q}_{\text{T}}, \overline{q}_{\text{p}}, \overline{u}, \overline{v}, \overline{w}), \quad (5)$$

where $B$ is a certain variable, $\frac{\partial \overline{B}}{\partial t}(h_{\text{L}}, q_{\text{T}}, q_{\text{p}}, u, v, w)$ is the coarse-grained high-resolution tendency of that variable due to the process, $\frac{\partial B}{\partial t}(\overline{h}_L, \overline{q}_T, \overline{q}_p, \overline{u}, \overline{v}, \overline{w})$ is the resolved tendency due to the process, and $\left(\frac{\partial \overline{B}}{\partial t}\right)^{\text{subgrid}}$ is the subgrid tendency due to the process. For example, the subgrid tendency of $h_{\text{L}}$ due to vertical advection is

$$\left(\frac{\partial \overline{h}_{\text{L}}}{\partial t}\right)^{\text{subgrid}}_{\text{vert. adv.}} = -\left(\frac{\partial \overline{w h_{\text{L}}}}{\partial z} - \frac{\partial \overline{w} \overline{h}_{\text{L}}}{\partial z}\right). \quad (6)$$

Subgrid and resolved contributions are defined in a similar way for the surface fluxes of $h_{\text{L}}$ and $q_{\text{T}}$.

The procedure of coarse graining and calculating the subgrid tendencies and subgrid surface fluxes was done offline in postprocessing. For each high-resolution snapshot, the coarse-grained fields, the instantaneous tendencies associated with different physical processes, and the surface fluxes were calculated. The coarse-grained fields were then used to calculate the instantaneous resolved tendencies of the different physical processes and the resolved surface fluxes. Finally the subgrid contributions were calculated. This procedure is more accurate compared to previous studies that calculated the tendencies using the difference between the prognostic variables over 3-h time steps[14,15]. Furthermore, this procedure allows us to calculate a different subgrid tendency for each physical process, which is necessary for the RF parameterization structure that we use.

**Choice of outputs for the RF parameterization.** The RF parameterization predicts the combined tendencies for the following processes: subgrid vertical advection of $h_{\text{L}}$, $q_{\text{T}}$, and $q_{\text{p}}$, subgrid cloud and precipitation microphysical tendencies included in $\left(\frac{\partial q_{\text{p}}}{\partial t}\right)_{\text{mic}}$, subgrid falling of precipitation and subgrid sedimentation of cloud ice, and the total radiative heating tendency (see below). The RF parameterization also predicts the coarse-grained turbulent diffusivity and the subgrid corrections to the surface fluxes of $h_{\text{L}}$ and $q_{\text{T}}$.

For radiation, the RF parameterization predicts the total radiative heating and not the subgrid part. The choice to predict the radiative heating tendency rather than predicting its subgrid correction was mainly motivated by the complexity of calculating subgrid radiative heating tendencies in postprocessing. Radiative heating is not predicted above 11.8 km since the RF has poor performance above this level in offline tests. Instead the SAM prediction for radiative heating is used at those levels. We checked that the results were not sensitive to the exact choice of cutoff level. Including the RF prediction for radiative heating at all stratospheric levels leads to a temperature drift in the stratosphere when RF-tend is implemented in SAM (a problem with temperatures in the stratosphere was also found in a previous study[12]), though tropospheric fields are still similar to the presented results. It is possible that due to weak troposphere-stratosphere coupling it is difficult to accurately predict the radiative heating tendency simultaneously in both the troposphere and the stratosphere. In future work, it might be beneficial to train different parameterizations for the stratosphere and troposphere.

The turbulent vertical diffusive flux for a thermodynamic or moisture variable $A$ is $F_{Az} = -D\frac{\partial A}{\partial z}$, where $D$ is the turbulent diffusivity for thermodynamic and moisture variables. We predict the coarse-grained turbulent diffusivity ($\overline{D}$) and apply it only to vertical diffusion of the thermodynamic and moisture variables (i.e., $h_{\text{L}}$, $q_{\text{T}}$, $q_{\text{p}}$). This is consistent with our general approach in which the RF parameterization only represents processes that act in the vertical and their effects on the thermodynamic and moisture variables. The approach of predicting the coarse-grained diffusivity has the advantage that it constrains the diffusive fluxes in the coarse model to be downgradient, unlike if we had predicted the tendency due to diffusion. This approach also had the advantage that the same diffusivity is applied to all thermodynamic and moisture variables, unlike if we had predicted the effective diffusivity based on coarse-grained fluxes and gradients for each variable separately. The coarse-grained diffusivity is not predicted above 5.7 km, and the diffusivity from SAM at coarse resolution is used instead for these levels.

Surface precipitation is not predicted separately by the RF parameterization but is rather diagnosed (including any surface sedimentation) as the sum of the resolved precipitation and the subgrid correction ($P_{\text{tot}}^{\text{subgrid}}(z=0) + S^{\text{subgrid}}(z=0)$) which is calculated from water conservation as

$$P_{\text{tot}}^{\text{subgrid}}(z=0) + S^{\text{subgrid}}(z=0) = -\int_0^\infty \left(q_{\text{p}}^{\text{subg-tend}} + q_{\text{T}}^{\text{subg-tend}}\right)\rho_0 \text{d}z. \quad (7)$$

**Training and implementation.** Before training the RFs, each output variable is standardized by removing the mean and rescaling to unit variance. For output variables with multiple vertical levels, the mean and variance are calculated across all levels used for that output variable.

We use 337.5 days of 3-hourly model output from the hi-res simulation to calculate the features and outputs of the RFs. This model output was divided into a training dataset, validation dataset and a test dataset. The training dataset was obtained from the first 270 days (80% of the data) of the hi-res simulation, the validation dataset was obtained from the following 33.75 days (10% of the data), and the test data was obtained from the last 33.75 days (10% of the data). After tuning the hyperparameters, we expanded the training dataset to include the validation dataset for use in the final training process of the RFs used in SAM.

To make the samples more independent, at each time step that was used, we randomly subsample atmospheric columns at each latitude. For coarse-graining factors of ×4, ×8 and ×16, we randomly select 10, 20 and 25 longitudes, respectively, at each latitude for every time step. For ×32, the amount of coarse-grained output is relatively limited and so we do not subsample. This results in test and validation dataset sizes of 972, 360 samples for ×4 and ×8, 607, 770 samples for ×16 and 218, 790 samples for ×32. The amount of training data used is one of the hyperparameters we tuned as described below.

To train the RFs, we use the RandomForestRegressor class from scikit-learn package[37] version 0.21.2. Different hyperparameters governing the learning process and complexity of the RFs may be tuned to improve performance. The most important hyperparameters that we tuned are the number of trees in each forest, the minimum of samples at each leaf node, and the number of training samples. Supplementary Fig. 8 shows the coefficient of determination ($R^2$) evaluated on the validation dataset for different combinations of hyperparameters. We stress that unlike standard supervised machine-learning tasks, higher accuracy on test data is not our only goal. We also want to have a fast RF since it will be called many times when used in a simulation, and we do not want to have an RF that is overly large in memory since it will need to be stored on each core (or possibly shared across all cores in a node). Based on a compromise between RF accuracy, memory demands and speed when the RF is implemented in SAM, for coarse-graining factors of ×4, ×8 and ×16 we chose 10 trees in each RF, a minimum of 20 samples in each leaf and 5,000,000 training samples. However, fewer training samples were available for ×32, and in order to have a similar size of RFs in this case, a minimum of seven samples in each leaf were taken.

Training typically takes less than an hour using 10 CPU cores. For x8, RF-tend is 0.75 GB and RF-diff is 0.20 GB when stored in netcdf format at single precision. We found that this size in memory did not pose a problem when running across multiple cores. We also emphasize that the RF parameterization can achieve similar accuracy at a smaller size. For example, we reduced the number of trees in RF-tend from 10 to 5 which reduces its size in memory by more than a factor of two to 0.35 GB without any noticeable difference in the results when it is implemented in SAM at coarse resolution. Furthermore, there are available techniques to reduce the memory needed to store RFs[38–40] in case memory becomes a limiting factor when using an RF parameterization in operational climate simulations with more degrees of freedom. Each RF was stored as a netcdf file, and routines to read in the netcdf files and to use the RFs to calculate outputs were added to SAM (using Fortran 90).

**Offline performance.** Offline performance is primarily evaluated using the coefficient of determination ($R^2$) as applied to the unscaled output variables in the test dataset. $R^2$ is plotted for outputs of the RF parameterization as a function of the latitude and pressure in Supplementary Fig. 9. For reference, the standard deviation of true outputs is plotted in Supplementary Fig. 10 and the mean of the true outputs is plotted in Supplementary Fig. 1. $R^2$ is generally higher in the lower and middle troposphere, though performance does vary across outputs. Generally, the RFs tend to underestimate the variance in predictions compared to the true variance, although less so for larger coarse-graining factors (Supplementary Fig. 11). $R^2$ for the different outputs (combining data from all vertical levels for a given output) at different coarse-graining factors are given in Supplementary Table 2, and corresponding values of the RMSE are given in Supplementary Table 3.

RF-tend is also able to accurately predict the instantaneous surface precipitation rate (Supplementary Fig. 3) with $R^2 = 0.99$ based on the test dataset for ×8. The predicted precipitation (including any surface sedimentation) is the sum of the resolved precipitation and the predicted subgrid correction ($P_{\text{tot}}^{\text{subgrid}}(z=0) + S^{\text{subgrid}}(z=0)$) which is calculated from Eq. (7).

To further investigate the offline performance at different grid spacing, we focus on a comparison between ×4 and ×32. We coarse grain the subgrid tendencies calculated and predicted at ×4 to the same grid as ×32 (referred to as ×4 → x32) such that they are on the same grid as the subgrid tendencies calculated and predicted at ×32. To do the coarse graining, it was necessary to make an alternative test dataset since the default test dataset is randomly subsampled in longitude for ×4. 100 snapshots from the hi-res simulation were used without subsampling in longitude. This results in an alternative test dataset size of 5,184,000 for the ×4 case and 81, 000 for the ×4 → x32 and ×32 cases. We find that calculating $R^2$ and RMSE from these 100 snapshots gives almost identical results compared to the test dataset that was used for model evaluation (compare Supplementary Tables 2 and 3 to 4). One snapshot from the alternative test dataset is shown in Fig. 4, and we also use the alternative test dataset for the results shown in Supplementary Fig. 5 and in Supplementary Table 4.

## Data availability

RF estimators and snapshots for different resolutions are available at osf.io (https://doi.org/10.17605/OSF.IO/36YPT). Additional data that support the findings of this study are available from the corresponding author upon request.

## Code availability

Associated code are available at osf.io (https://doi.org/10.17605/OSF.IO/36YPT).

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

## Acknowledgements

The authors thank Bill Boos for providing the output from the high-resolution simulation, and Daniel Koll, Nick Lutsko and Chris Hill for helpful discussions. We acknowledge high-performance computing support from Cheyenne (doi:10.5065/D6RX99HX) provided by NCAR's Computational and Information Systems Laboratory, sponsored by the National Science Foundation. We acknowledge support from the MIT Environmental Solutions Initiative, the EAPS Houghton-Lorenz postdoctoral fellowship, and NSF AGS-1552195.

## Author contributions

J.Y. and P.O. designed the research. J.Y. performed numerical simulations. J.Y. and P.O. wrote the manuscript.

## Competing interests

The authors declares no competing interests.
