## [Peer Review File · Nature Communications]

Reviewers' comments:

Reviewer #1 (Remarks to the Author):

The question of what horizontal scale regimes of deep convection are vs. aren't parameterizable is of broad and long-standing interest. The authors have devised a clever new attack on this through analyzing the sensitivities of an unprecedented modeling approach in which machine learning of the deterministically predictable component of high-resolution atmospheric simulation data is done at varying coarse-grained resolutions. Despite the fact that this is done only in a limited basic state, and in only one such modeling framework, the high-profile nature of the overarching question about "parameterizability", and the fact that this is the first time the question has been addressed with such tools -- which themselves are updated in many ways that make important technical progress -- is of appropriate impact for N. Comm.

My main critique of the article is that its main message can appear to be about the incremental success in the surrogate modeling technique (e.g. abstract is exclusively about this, and the order of take-homes in concluding discussion suggests it is of higher emphasis). While important, these details are not of sufficiently broad interest to the N. Comm. audience. This is not the first paper to attempt to fit coarse-grained cloud-resolving model (CRM) physics with a machine learning based surrogate model. On the one hand, it is an unusually high quality and technically thorough attempt, insofar as the authors have gone to great lengths to respect the distinctions between prognostic vs. diagnostic variables in the CRM that they are using as a training data set. But on the other hand, some key tests of generalizability beyond reproducing one limited basic state have not been attempted, such that one cannot yet claim a step change in convection-permitting operational global climate simulation.

Nonetheless what convinces me the paper should still be of interest to the N. Comm. readership is what the authors ultimately do with their new, hard-won ML modeling framework--assessing at what horizontal scales parameterization itself does vs. doesn't make sense. With a bit of major revising to help this shine as the main motivation and take-home message, I would recommend the paper be published in N. Comm.

Major comments:

1. [EITHER]: Please revise the abstract, introduction and conclusions to better set the stage about the culminating question that is answered about what scales are parameterizable in ways that the average N. Comm. Reader will be equipped to appreciate. In this context, it would help to clarify that the majority of text devoted to technical advances in how to do the ML surrogate modeling right are a means to this scientific end, rather than the end & message themselves.

2. [OR] To better substantiate an alternate message of a broadly impactful breakthrough in ML-based climate simulation please include tests of generalizability by e.g. moving the SST maximum off the equator, which seems quickly approachable with the existing modeling framework while avoiding any issues of going out-of-sample on inputs in ways that would challenge a RF. Such tests would helpfully ward off any concerns that the RF parameterization has indirectly memorized geographic details of its training climate through information such as the distance from equator or through the enforced hemispheric symmetry of the meridional wind inputs. Also, if this is meant to be the main message some advocacy and added nuance about what the broader ML parameterization community should pursue in light of these results vs. previous ones would be appropriate for this journal.

Minor comments:

1. The authors could brag a little more about the fact that they are the first to have attempted the ML parameterization exercise in a way that deeply respects the model equations of SAM, including taking great care to only predict what is prognostic. This would help the unfamiliar reader appreciate a technically admirable dimension of this paper, and help add context relative to previous results.

2. In addition to spatial scale, can you comment on the temporal scales of ML parameterization and how this result situates in the context of e.g. Rasp et al. and Bretherton & Brenowitz? Is there ambiguity between the time separation of the training data (3-hour) and the time scale at which the parameterization is implemented in SAM?

3. Does the fact that the diurnal cycle is avoided make the training problem simpler than past attempts in ways that further nuance the comparisons?

4. SI, L45: I think Rasp et al. also trained on instances of SAM, albeit embedded in a host GCM.

5. SI, L81-84 – this text makes it sound like the sub-grid tendencies were defined using a two-stage process where coarse grained inputs were fed into the SAM model to produce resulting resolved scale tendencies... but my impression from the subsequent equations (Eq. S6) is that the authors actually just horizontally averaged the high-resolution data directly offline. Please clarify the language here, either way.

6. SI, L103-4 – what is it about stratospheric radiative heating and turbulent diffusivity above 5.7 km that makes it hard to fit with a RF? Do the authors have any intuition into this issue? Is this a limitation of the RF approach to climate simulation or is there an obvious fix? Some comments about “where the work is leading” seem appropriate for N. Comm – beyond the exciting success to also highlight the challenges; for instance, the fact that some bits have to be patched with the explicit solution speak to broader issues of robustness in need of community attention, which this article could help galvanize.

7. L110-118 – Bravo! This is a very crafty and sensible way to avoid inconsistency in fitting the SGS diffusion of moisture vs. temperature, and to enforce the downgradient constraint.

8. SI, L117-118 – can you please elaborate on this?

9. SI, L132: Is there any vulnerability that by using distance from equator as an input the parameterization may be memorizing the basic state that it was trained on? Allowing this as an input seems incongruous with the eventual target of an operational parameterization, which should be able to generalize to new basic states via direct (rather than proxy) information about relevant boundary conditions. Does the parameterization work when the SST, surface albedo and solar insolation are used instead (and if so, can it generalize to new basic states when used online; see major comment #3)?

10. SI, L147-151: This decision could also appear prone to over-fitting or baking in the basic state in ways that would inhibit generalizability for broader climate simulation. Please elaborate or clarify.

11. SI, Fig. S2 – please add the corresponding pressure-lat sections of time variance for each of the fields for context, assuming that that some of the geographic structures in R^2 will be controlled by the baseline target variance.

12. SI, L222: never predicts  always predicts?

13. The discussion about memory limits of RF networks is very interesting. It is good to know that for the diversity of regimes that need to be learned in this idealized simulation of a slice of aquaplanet, 0.5 Gb is enough. Do the authors have any intuition on how much more complexity / memory will be needed in the limit of operational climate simulation with more DOF?

Reviewer #2 (Remarks to the Author):

This paper describes the use of machine learning, specifically random forests, to represent small scale processes in climate models. This study logically builds upon previous research in this area and, for the first time, achieves a stable simulation with a machine learning model trained on coarse-grained high-resolution data. The paper is well written and the methods and results are presented in a logical way. All analyses are statistically sound. I believe that this paper will be of great interest to a broad audience. I recommend to accept this paper with minor revisions. My comments are listed below.

Best,
Stephan Rasp

Key comments:

Data and code availability: As far as I can see the paper currently contains no statement about data and code availability, which I think is required by Nature journals. I understand that the raw dataset is likely very large and hosting the data is not easily possible. However, it would be great if the authors could share a small selection of the data (just a few time steps) on Figshare or Zenodo. I think this could be helpful to other researchers in this field. While there is no requirement to share code, I would strongly encourage the authors to share their code. Even if the code is messy, just looking at which functions were used for coarse-graining, etc. would be very helpful for the few people, like myself, who are also actively working on this topic. The large share of reader will never look at the code (so it doesn't matter if it's not pretty) but the few researchers who will, would be happy to see it even if it is not well documented.

Title: One could argue that the title overstates what is achieved in the paper. I wholeheartedly agree that this study is a significant step forwards but I don't think that at this stage it actually "improves" climate simulations. The model is still idealized and does not compare to observations and other realistic climate models. I understand that titles like these are common at journals like NatComms but I would prefer a more accurate title that highlights the achievements of the paper, namely stable simulations using coarse-grained high-res data.

Final discussion: I think that the discussion could be enhanced. In particular, there is no mention about the RF vs NN question. I think that a sentence or two highlighting the advantage of RFs over NNs is certainly warranted since, to my knowledge (having last talked to Noah and Chris a couple of

months ago), NNs are still struggling with stable simulations. Plus, there seems to be no real reason for using NNs, since their advantages (differentiability, complex architectures) have not been used for parameterizations. Another missing discussion, in my opinion, is the difference of your coarse-graining method with the one of Brenowitz and Bretherton. I think this would be of great interest to many readers. This could go at the end or when describing the coarse-graining, which brings us to the next point. For example, one advantage of your method is that it does not require you to do any temporal or horizontal differencing.

Main text description of coarse-graining and RFs (line 90ff): A lot of the details about the method are deferred to the supplement. I believe that it would make sense to add more detail to the main text, as currently it is very high-level. In particular, I think the text could benefit from a more in-depth coverage of: a) Why are two RFs necessary? This is actually not entirely clear to me even with the Supplement. You mention that RF2 has different inputs (upper-levels cut off) to save time but would it really be computationally impossible to have a single RF? b) What are the inputs/outputs? I think moving the input and output vectors from the supplement (e.g. supp. line 132) to the main text would make the text easier to understand. c) What is diffusivity? Where does what you predict show up in Eqs. S1-S3? This point could be covered in the Supplement only.

Using $|y|$ as input: The distance to the equator was used as a proxy for some latitude-dependent variables. I wonder, however, if there is a slight potential for "cheating". The data has a very strong latitudinal structure, so the RF could simply learn that structure rather than basing its predictions upon the actual SST, etc. I wonder what would happen if SST, albedo and insolation were directly fed into the RF. Most likely the differences would be small but maybe it would be good to at least add a sentence discussing this if the actual experiments are too difficult to run. For real geography cases, $|y|$ is not sufficient anyway.

Small comments and typos:

Line 35: "A NN" instead of "An NN"

Line 62: "ocean" Maybe it would be clearer to explicitly mention the word "aquaplanet" which has been commonly used.

Line 106: "necessary to include multiple time steps" This also would not be possible using a RF since it's not differentiable, right? I also found the following sentence a little unclear: "different approach", "different machine learning algorithm". I think it would be clearer if you explicitly names what these different approaches are (see my comment on mentioning the differences in coarse-graining above).

Line 153: At the beginning of this section you ask the question whether there is a scale particularly suitable for parameterization. So what is your conclusion based on your results? Is it $8x = 100\text{km}$? Maybe add a sentence or two of discussion.

Reference 17. My paper now has a slightly different, hopefully less confusing, title: "Coupled online learning as a way to tackle instabilities and biases in neural network parameterizations"

Supplement

Line 33: What height is the model top?

Line 86: Just out of curiosity, how did you technically compute the resolved tendencies? Did you coarse grain your high-res fields and then feed them to low-res SAM to compute the tendencies?

Line 103: "not predicted above". I have several questions about the cut-off of radiative predictions. First, does the switch from RF to SAM lead to any noticeable discontinuities? Second, why does including the RF heating rates lead to a drift? It would be great if you could dig a little deeper into this topic and maybe discover some mechanism. In our PNAS paper we also had strong biases in the stratosphere. I was never able to find out exactly why this was happening. But if you are having similar issues, maybe there is a more systematic cause?

Line 199: It would be very interesting if you stated somewhere how you implemented the RF in SAM. Did you hardcode the RF in Fortran or did you interface with Python?

Line 249: Energy conservation error: It might be helpful to state somewhere the mean absolute magnitude of the vertically integrated tendency as a reference for the residual values. Without context it is hard to appreciate how accurate the RF is.

Line 269: "timestep" Did you try running this parameterization at a coarser time step?

Line 292: "Third, ..." I personally would like a more in-depth discussion of the instabilities. As far as I understand Noah's instabilities can be traced back to a wrongly reversed causality, where upper level Q perturbations lead to strong tropospheric heating. So you suspect the same mechanism? Why does adding q_p prevent this?

Table S2: Also show lat-z mean plots of R2.

Reviewer #3 (Remarks to the Author):

Please see attached review. I am happy to waive my right to anonymity and I have signed my review in the attached pdf.

Review of ‘Use of machine learning to improve simulations of climate’

by Janni Yuval and Paul O’Gorman

Nature Communications manuscript number NCOMMS-19-42290-T

Recommendation: Minor Revision

Reviewer Name: Penelope Maher

1 Summary of the Review

The authors use 3D high resolution model output from SAM to train machine learning parameterisations using the Random Forests technique. In implementing their machine learned parameterizations in online testing, the authors are able to create realistic and stable simulations of the climate. These results are novel, interesting and worthy of publication in Nature Communications. The manuscript is very well written and in a style consistent with the journal. The figures are also very well presented (especially Fig 1.). I would be happy to read the manuscript a second time should it be needed.

A relevant caveat for my review of this manuscript is to state that I have not implemented any ML techniques. As such, I can not comment on the implementation choices or the subtleties of the method. However, I am experienced with modelling convection and model development more generally and it is this experience on which my review is based.

I have listed a few minor comments and a longer list of clarification items. In the majority of cases, my comments are from the perspective of helping non-ML experts read the manuscript and interpret the significance of the results. I have also included a list of editorial comments. There is no need to respond to any of the editorial comments in your response to reviewers. The majority of these editorial comments are suggestions and can be changed at the authors discretion.

2 Minor Comments

1. Abstract - I think you should mention these are online parameterisations right at the start. This is an important piece of info.
2. The manuscript would benefit from a sentence or two explaining the RF technique (or at a minimum pointing the reader to some refs), citing the code used if possible, and a longer introduction to the RF techniques in the supplementary.
3. L155-162 I think the discussion could be longer. I feel you undersell the significance and importance of your simulations.
4. Sup L34-35: You mention that the time step is halved when the model is unstable. Is the model often unstable? At what resolutions was this a problem? Was it only early in the simulations or often crash later on?
5. Sup L39-42 The hypohydrostatic rescaling is usually applied to reduce the resource requirement to run these simulations. Given the ML parameterisation you develop is much faster than traditional parameterizations, why use this rescaling? Was this needed to make more (or longer) training data sets?

3 Clarifying Comments

3.1 Abstract

1. L8: I think you should mention the model domain or that it is idealised to avoid confusion with 3D global high-res GCMs (will the full complexity of such simulations).

3.2 Introduction

1. Is ‘learn’ and ‘train’ the same thing? They are used interchangeably in places (eg L 5, 7, 12, 26, 56). In my mind we use training data to train a ML scheme and that ‘learning’ is the resulting statistical model that is implemented as the scheme.
2. I think you should state early more clearly how this study is similar and different to O’Gorman and Dwyer (2018). Or at a minimum state that the method is the same but the implementation is different. This will help make the new results of your study clearer.
3. L28-31 I was under the impression that ML in warmer climates fail to simulate realistic climates (when trained on historical data). Do you mean that training should include warmer climates and then it is trivial? Would you need to include a very large temperate range in your training data?

3.3 Results

1. Did you train the data on the same system that SAM was run on? Or was the training done on GPUs?
2. The double ITCZ in high-res simulations vs single ITZC in the x8 runs is very interesting. Has this been seen in SAM before?
3. L126-129 I am confused about the use of the alternative RF parameterization and the different treatment of q_p . Is this only for speed or are there other advantages? If this has similar results and is cheaper, then why not is it instead?
4. Fig 3: I would reorder the subplots so that the second row is the q_T tendency contours (the subplots c and e would also benefit from units or color bar table or subheading) and the third row is P. If this is not possible, then I suggest moving d and f to a new plot so that R^2 and zonal precip are separated. If you are aiming for a three fig paper, then perhaps put the Precip plots in fig 2?

3.4 Supplementary

1. In equ S1 and S2 there are large-scale prescribed terms. Could you add a comment in the appendix about why these are not in your formulation? Could you also add a comment on why you included the sublimation term (S) in equation S2.
2. Sup L26-29 Could you explain what the partition function (w) is? I assume this means how the water species are separated but I am not sure.
3. Sup L34. Is 24 sec the default for SAM or the default time step for your study?
4. Sup L47-49. What is the significance of saving snapshots every 3 hours? Is hourly to data heavy and 6 hourly not helpful? I don’t see the significance of 3 hours and I assume there is no sensitivity to this frequency given it is a snapshot (given the simulations are stable).
5. Sup L127-127: could the local time/space assumption be relaxed so that the system could have memory or organisation?
6. Sup L 160: Why is the variability important but not the mean? What then sets the mean?
7. Sup L 170-172: Might be worth mentioning in here that low correlations between samples is the goal - to non ML people low correlations would read as a bad thing.
8. Sup L 178: Could you explain what hyperparameters are used before describing their tuning?
9. Sup L 182: Could you add that ‘R’ is the correlation to help people interpret R^2 (might be good to say this in the manuscript as well)?
10. Sup L269: what do you mean by a ‘fast variable’?
11. Equ S11: Could you check the sign of the last term please? The P_{tot} looks fine but should the other two terms have opposite sign?

4 Editorial comments

The following editorial comments are suggestions only. I am happy for the authors to execute their own discretion on all of these. There is no need to respond to any of these points in your response to reviewers.

4.1 Introduction

1. L18 Suggest replacing ‘main’ with ‘leading’.
2. L19 Suggest you add ‘future’ before ‘projections’.
3. L24 It might also be worth noting that these simulations can’t be run for the many different scenarios that are needed for model intercomparison studies.
4. L27-32 (and elsewhere): Are the ML and NN acronyms needed? The RF acronym is probably fine as it is used regularly and in figures but ML and NN are not used enough in my opinion to be used as acronyms.
5. There were a number of times when ‘an’ was used in front of an acronym when I think ‘a’ should be used. I am not skilled at using indefinite articles so I will leave this to the authors to decide.
6. There are a number of references that are in the supplementary material that are not in the main of the paper. I suggest adding you cite all the relevant refs in the main manuscript where relevant.
7. L34-35 suggest replacing ‘blow ups’ with ‘numerical instabilities’ or something similar.
8. L49-50 is there a more plain english word you could use instead of ‘convex hull’?
9. L52 suggest replacing ‘conventional’ with ‘traditional’

4.2 Results

1. L55-59 Suggest breaking this down into at least 2 sentences as there is too much here to digest as one sentence.
2. L56 It would be good to state here or sooner what course-graining is and why is it needed. I would also change ‘course resolution’ to ‘low resolution’ to avoid confusion here.
3. L110 suggest replacing ‘(text S3)’ with ‘(see section S3)’ (likewise L126)
4. L111-113 I had to read this sentence multiple times before I understood what you meant. Suggest you flip the order of information: ‘The x8 simulation with traditional parameterization was used to provide the initial conditions for the RF parameterization simulation at 96km grid spacing (8x-RF) which was trained using the high-res simulation’ or something similar.
5. L115 replace ‘it runs stably’ with ‘is stable’

4.3 Supplementary material

1. Sup L34. It might be helpful to say what the pressure at the model top is.
2. Sup L39-42 It might be worth adding in the text that this is similar to DARE and cite Pauluis et al. (2006) if you feel it is relevant to this section.
3. Equ S4: Is ‘N’ the standard notation for the course graining factor? If it is not, then I suggest another variable letter to avoid any confusion with the Brunt–Väisälä frequency (I appreciate it is unrelated in this context but it was the first thing I thought of when I saw this formula).
4. Sup L72-74: I found these two sentences confusing. Maybe a table would help or perhaps a rewrite?
5. Sup L75: Sentence fragment.
6. Sup L126: It might be a helpful to say here that 2 RFs are needed due to how diffusion is treated in the model. Then when you introduce the two RF’s the reader is prepared (on first reading I spend a while thinking why you need 2 RFs?)
7. Sup 133: Please add in brackets where the components of the feature number originates? I assume 48 is the vertical levels but I am not sure where the 3 comes from (is it because there are three tendencies?) or why +1 is needed. Is outputs = features -1 a general rule?

8. Sup L139 (and elsewhere) suggest replacing RF-diff with RF-diffuse (to avoid any mistakes in thinking this is a ‘difference’).
9. Sup L157 and S7 (and elsewhere): suggest removing ‘($z = 0$)’ and replacing ‘which is calculated’ with ‘which is calculated at the surface’. If you do this, I then suggest to spell out ‘subg’ as ‘subgrid’ so it will look better.
10. Sup LS7: Suggest removing ρ_0 out of the integrand and placing after the negative sign.
11. Sup L172: Why 10, 20, and 25 samples? Trial and error? Were they sampled with or without replacement?
12. Sup L263: I suggest you start by saying why not having precipitating water as a variable is a good thing. You say this toward the end of the section to some degree but it would be helpful to say it upfront.
13. Sup L275: the ref seems unnecessary at this point.
14. Sup L275-276: suggest remove one of the two uses of ‘vertical’.
15. I suggest moving Fig 6-7 and table S2 to the start of the supplementary material. Then followed by the methods section. I would also integrate the figures with the text for easier reading.

References

- Pauluis, O., Frierson, D. M. W., Garner, S. T., Held, I. M., and Vallis, G. K. (2006). The hypohydrostatic rescaling and its impacts on modeling of atmospheric convection. *Theoretical and Computational Fluid Dynamics*, 20(5):485–499.

Response to reviewer 1

We thank reviewer 1 for their helpful comments which have improved the paper.

The question of what horizontal scale regimes of deep convection are vs. aren't parameterizable is of broad and long-standing interest. The authors have devised a clever new attack on this through analyzing the sensitivities of an unprecedented modeling approach in which machine learning of the deterministically predictable component of high-resolution atmospheric simulation data is done at varying coarse-grained resolutions. Despite the fact that this is done only in a limited basic state, and in only one such modeling framework, the high-profile nature of the overarching question about 'parameterizability', and the fact that this is the first time the question has been addressed with such tools - which themselves are updated in many ways that make important technical progress - is of appropriate impact for N. Comm.

My main critique of the article is that its main message can appear to be about the incremental success in the surrogate modeling technique (e.g. abstract is exclusively about this, and the order of take-homes in concluding discussion suggests it is of higher emphasis). While important, these details are not of sufficiently broad interest to the N. Comm. audience. This is not the first paper to attempt to fit coarse-grained cloud-resolving model (CRM) physics with a machine learning based surrogate model. On the one hand, it is an unusually high quality and technically thorough attempt, insofar as the authors have gone to great lengths to respect the distinctions between prognostic vs. diagnostic variables in the CRM that they are using as a training data set. But on the other hand, some key tests of generalizability beyond reproducing one limited basic state have not been attempted, such that one cannot yet claim a step change in convection-permitting operational global climate simulation.

Nonetheless what convinces me the paper should still be of interest to the N. Comm. readership is what the authors ultimately do with their new, hard-won ML modeling framework - assessing at what horizontal scales parameterization itself does vs. doesn't make sense. With a bit of major revising to help this shine as the main motivation and take-home message, I would recommend the paper be published in N. Comm.

As described in detail below, we have revised the paper to emphasize our results on performance as a function of horizontal length scale. We have also added some generalization tests and other tests to demonstrate the robustness of the RF parameterization.

Major comments:

1. [EITHER]: Please revise the abstract, introduction and conclusions to better set the stage about the culminating question that is answered about what scales are parameterizable in ways that the average N. Comm. Reader will be equipped to appreciate. In this context, it would help to clarify that the majority of text devoted to technical advances in how to do the ML surrogate modeling right are a means to this scientific end, rather than the end and message themselves.

In response to this comment we added text to the introduction (lines 37-46, 76-79) and discussion section (lines 269-280) on the issue of parameterization across length scales. Furthermore, we now highlight the motivation to study this point in the abstract: "...and their

performance for different grid spacings has not yet been investigated” and we also say that “Retraining for different coarse-graining factors shows the parameterization performs best at smaller grid spacings. Our results yield insights into parameterization performance across length scales,...” and we have removed some text from the abstract about the performance of the parameterization to make space for this new emphasis. We didn’t include the issue of grid spacing in the title because Reviewer 2 had a different request for the title of the paper and the title has a word limit. We also added additional novel results on how best to compare offline performance across grid spacings (see Fig. 4, text within the manuscript on lines 204-226 and Fig. S7).

2. [OR] To better substantiate an alternate message of a broadly impactful breakthrough in ML-based climate simulation please include tests of generalizability by e.g. moving the SST maximum off the equator, which seems quickly approachable with the existing modeling framework while avoiding any issues of going out-of-sample on inputs in ways that would challenge a RF. Such tests would helpfully ward off any concerns that the RF parameterization has indirectly memorized geographic details of its training climate through information such as the distance from equator or through the enforced hemispheric symmetry of the meridional wind inputs. Also, if this is meant to be the main message some advocacy and added nuance about what the broader ML parameterization community should pursue in light of these results vs. previous ones would be appropriate for this journal.

The reviewer indicated that adding generalization tests was not necessary if the paper was changed to highlight the dependence of parameterization on grid spacing (which we have done as discussed above). Nonetheless, given the reviewer’s comments we thought it was important to also add a new section entitled "Robustness of the RF parameterization" in which we show tests of generalization by excluding latitude bands from training and we also added other tests of robustness:

- In order to demonstrate that the RF parameterization could succeed in performing well in circulation regimes or latitudes that it did not train on, we re-trained while excluding certain latitude bands from the training process. We ran 3 additional simulations using an RF parameterization that did not train on certain latitude bands (in both hemispheres) and demonstrate that the results of these simulations are similar to the simulation that ran with RF-tend that was trained on the whole domain (see lines 236-265, the new figure 5 and figure S11). For example, excluding latitude bands that contain both ITCZs from the training does not affect the performance. We note that we can exclude wider latitude bands and still generalize well in the tropics, but to exclude wider latitude bands in midlatitudes would require changing features (e.g. relative humidity instead of mixing ratio) and this could be explored in future work. Note also that we did not follow the suggestion of the reviewer to test generalization using a simulation with an SST peak off the equator because this would require running one or more new high-resolution simulations and because we think this kind of generalization test is best performed using simulations with a full seasonal cycle (which we will do in future work).
- To demonstrate that the RF is not memorizing geographic details of its training climate

Figure 1: Zonal- and time-mean precipitation and for (a) the x8-RF standard simulation (red) and the x8-RF simulation without the distance from equator as a feature (dotted black), and (b) the x8-RF standard simulation (red) and the x8RF without using the mapping v to $-v$ in the southern hemisphere (dotted black)

through the distance-from-equator feature we have re-trained the parameterization without using the distance from equator as a feature and found that results are almost identical (see table S1 for the offline results). Furthermore, we plot here the results of the mean precipitation from this simulation (Fig. 1a in this response). This is reassuring since much of the subgrid physics (vertical advection, cloud and precipitation microphysics, longwave cooling) does not depend on or relatively weakly depends on the variables that $|y|$ is a proxy for (in the case of longwave cooling, the RF can use temperature at the lowest model level as a proxy for SST).

- To demonstrate that the RF is not relying on the transformation v to $-v$ across hemispheres used in training to respect the antisymmetry of v in the simulations, we show the mean precipitation from a simulation in which the RF parameterization was trained without the mapping v to $-v$ in the southern hemisphere. The online results (as well as the offline results) are almost identical to the standard x8-RF Fig. 1b) as discussed on lines 167-169 of the Supplementary Information.

We have also expanded the discussion section to cover the bigger picture of how our results compare to previous results and what should be done in future (e.g. how to compare across grid spacing, comparing NN versus RF on one dataset).

Minor comments:

1. The authors could brag a little more about the fact that they are the first to have attempted the ML parameterization exercise in a way that deeply respects the model equations of SAM, including taking great care to only predict what is prognostic. This would

help the unfamiliar reader appreciate a technically admirable dimension of this paper, and help add context relative to previous results.

In response to this comment we added a sentence to the discussion section (lines 284-287) that highlights this point.

2. In addition to spatial scale, can you comment on the temporal scales of ML parameterization and how this result situates in the context of e.g. Rasp et al. and Bretherton and Brenowitz? Is there ambiguity between the time separation of the training data (3-hour) and the time scale at which the parameterization is implemented in SAM?

During the training process we used snapshots from a high-resolution simulation in order to calculate the *instantaneous tendencies* of the prognostic variables in SAM. The training data was based on snapshots that were saved every 3 hours but this timescale doesn't affect the instantaneous tendencies that we train on. We use the calculated instantaneous tendencies as the output that the random forest predicts, and they are multiplied by the timestep of the coarse model to give the change in prognostic variables. By contrast, Brenowitz and Bretherton (2018) approximated the tendencies using the difference between the prognostic fields over interval of 3-hours (which was the output frequency of their high-resolution model). This is very different from our approach where we calculated the instantaneous sub-grid tendencies of each physical process separately. In Rasp et al. (2018) the aim of the ML parameterization was to mimic a superparameterization, where the time step in their (large scale model) was 30 min, and therefore the sub-grid tendencies that they predicted were on time steps of 30 minutes (which was the time step of the large scale model).

To clarify this issue, we added a sentence to lines 282-284 of the discussion section to make clear how our approach differs from Brenowitz and Bretherton. We also added text to the Supplementary Information explaining in more detail how we calculated the subgrid tendencies and highlighting the differences from Brenowitz and Bretherton (2018) (lines 103-112 of the Supplementary Information).

3. Does the fact that the diurnal cycle is avoided make the training problem simpler than past attempts in ways that further nuance the comparisons?

The fact that the diurnal cycle is not included in the simulations may make it easier for the training process although this is likely not a major issue since we prescribe SST and do not include land. We intend to investigate this issue in the near future and we now say it should be explored in future work in the description of the model on lines 41-42 of the Supplementary Information.

4.SI, L45: I think Rasp et al. also trained on instances of SAM, albeit embedded in a host GCM.

We added a citation to Rasp et al at the end of this sentence (line 48 of the Supplementary Information) to make clear that it was one of the studies that used SAM.

5. SI, L81-84 - this text makes it sound like the sub-grid tendencies were defined using a two-stage process where coarse grained inputs were fed into the SAM model to produce resulting resolved scale tendencies. but my impression from the subsequent equations (Eq. S6) is that the authors actually just horizontally averaged the high-resolution data directly offline. Please clarify the language here, either way.

Figure 2: The zonal- and time-mean coarse-grained (x8) turbulent diffusivity from the hi-res simulation as a function of pressure and latitude (upper panel) and the coefficient of determination (R^2) for an alternative version of RF-diff that predicts the diffusivity at all levels.

The procedure of producing the resolved tendencies and fields was done offline in a two-stage process. Namely, for every high resolution snapshot, we calculated the coarse-grained fields and fed them into a script that calculates the resolved tendencies and fields. This is now clarified in the Supplementary Information on lines 103-112.

6. SI, L103-4 - what is it about stratospheric radiative heating and turbulent diffusivity above 5.7 km that makes it hard to fit with a RF? Do the authors have any intuition into this issue? Is this a limitation of the RF approach to climate simulation or is there an obvious fix? Some comments about 'where the work is leading' seem appropriate for N. Comm - beyond the exciting success to also highlight the challenges; for instance, the fact that some bits have to be patched with the explicit solution speak to broader issues of robustness in need of community attention, which this article could help galvanize.

The diffusivity cut-off at 5.7km was used in order to reduce the computation time (see lines 146-151 of the Supplementary Information) and not because it was difficult to predict the diffusivity above this level. We introduced the cut-off because the diffusivity is generally small at higher levels (upper panel of Fig. 2 in this response to reviewers) with some small exceptions. To verify that we could predict the diffusivity reasonably well in all levels we re-trained RF-diff to predict all diffusivity levels (lower panel of Fig. 2 in this response to reviewers). Online performance when all levels are included for RF-diff is similar to what is presented in the paper (not shown).

The inability to predict accurately the stratospheric radiative heating might be related to the weak coupling between the stratosphere and troposphere as we now mention on lines 132-135. It is possible that two different RF algorithms, one trained on tropospheric levels and one trained on stratospheric levels, might be able to solve this issue as we now mention on lines 130-133 in the Supplementary Information. In response to this comment we added text to the discussion section that mentions this as one of the future technical challenges in ML parameterization (lines 299-300).

7. L110-118 - Bravo! This is a very crafty and sensible way to avoid inconsistency in fitting the SGS diffusion of moisture vs. temperature, and to enforce the downgradient constraint.

Thanks! In response to this comment we mention in the main text (lines 148-149) that the parameterization makes sure that turbulent diffusion is downgradient.

8. SI, L117-118 - can you please elaborate on this?

In order to keep the RFs affecting only vertical process that are directly related to the thermodynamic and moisture variables, we used the RF predicted diffusivity to only to the vertical diffusion of h_L , q_t and q_t . Namely, we did not change the horizontal diffusivity which was calculated by SAM or the vertical diffusivity used for the momentum variables. This is now clarified in the text on lines 134-145 in the SI

9. SI, L132: Is there any vulnerability that by using distance from equator as an input the parameterization may be memorizing the basic state that it was trained on? Allowing this as an input seems incongruous with the eventual target of an operational parameterization, which should be able to generalize to new basic states via direct (rather than proxy) information about relevant boundary conditions. Does the parameterization work when the SST, surface albedo and solar insolation are used instead (and if so, can it generalize to new basic states when used online; see major comment 3)?

We agree with the reviewer that in case we want to use the same algorithm in a less symmetric setup, we should train the RF using the relevant variables (SST, surface albedo and solar insolation) as features rather than distance to the equator. In the context of the current work, since RF is a decision tree based algorithm, where in each split of the trees a cut-point is chosen to divide the data to two different sub-data sets, using the distance from the equator or the SST, surface albedo and solar insolation is equivalent. In response to this comment, we added text to the main paper explaining this point (lines 141-145). We have also added a section to the manuscript on robustness of the RF parameterization that shows that not including the distance to the equator doesn't strongly affect our results (see also Fig. 1 in this response to reviewers).

10. SI, L147-151: This decision could also appear prone to over-fitting or baking in the basic state in ways that would inhibit generalizability for broader climate simulation. Please elaborate or clarify.

To verify that this transformation of v (which is motivated by the physics of the problem) did not affect the results, we re-trained the RF parameterization using v as a feature without the transformation $v \rightarrow -v$ in the southern hemisphere. The resulting offline performance is unchanged as now shown in Table S1, and the online results are also unchanged as shown

in Fig. 1b in this response to reviewers.

11. SI, Fig. S2 - please add the corresponding pressure-lat sections of time variance for each of the fields for context, assuming that some of the geographic structures in R^2 will be controlled by the baseline target variance.

In response to this comment we added to the Supplementary Information a figure (S3) showing the standard deviation of all outputs.

12. SI, L222: never predicts \rightarrow always predicts?

Corrected on line 262

13. The discussion about memory limits of RF networks is very interesting. It is good to know that for the diversity of regimes that need to be learned in this idealized simulation of a slice of aquaplanet, 0.5 Gb is enough. Do the authors have any intuition on how much more complexity / memory will be needed in the limit of operational climate simulation with more DOF?

It is difficult to estimate how much more memory would be needed for a RF to be used in operational climate simulations without actually trying to learn such parameterizations. Degrees of freedom (DOF) such as land, topography, diurnal cycle (for radiative heating) and others may increase the memory demands. In response to this comment, we added text and citations to the SI regarding the literature on compressing or reducing the memory requirements of RFs which could be helpful in an operational context (lines 224-227 in the SI)

References

- Brenowitz, N. D., and C. S. Bretherton, 2018: Prognostic validation of a neural network unified physics parameterization. *Geophys. Res. Lett.*, **45** (12), 6289–6298.
- Rasp, S., M. S. Pritchard, and P. Gentine, 2018: Deep learning to represent subgrid processes in climate models. *Proc. Natl. Acad. Sci. U.S.A.*, **115**, 9684–9689.

Response to reviewer 2

This paper describes the use of machine learning, specifically random forests, to represent small scale processes in climate models. This study logically builds upon previous research in this area and, for the first time, achieves a stable simulation with a machine learning model trained on coarse-grained high-resolution data. The paper is well written and the methods and results are presented in a logical way. All analyses are statistically sound. I believe that this paper will be of great interest to a broad audience. I recommend to accept this paper with minor revisions. My comments are listed below.

Best, Stephan Rasp

We thank the reviewer for their careful reading and the useful comments. Please see our reply to all the comments below.

Key comments:

Data and code availability: As far as I can see the paper currently contains no statement about data and code availability, which I think is required by Nature journals. I understand that the raw dataset is likely very large and hosting the data is not easily possible. However, it would be great if the authors could share a small selection of the data (just a few time steps) on Figshare or Zenodo. I think this could be helpful to other researchers in this field. While there is no requirement to share code, I would strongly encourage the authors to share their code. Even if the code is messy, just looking at which functions were used for coarse-graining, etc. would be very helpful for the few people, like myself, who are also actively working on this topic. The large share of reader will never look at the code (so it doesn't matter if it's not pretty) but the few researchers who will, would be happy to see it even if it is not well documented.

In response to this comment we have uploaded to Open Science Framework (doi: 10.17605/OSF.IO/36YH) the code we used for calculating the subgrid tendencies and SAM code we used for the simulations. We are also in the process of uploading (there is a problem with uploading large files at OSF at the moment) some of the training data, the RFs used in x8 simulations and some snapshots from the high-resolution output as well as the processed output (for different coarse-graining factors).

Title: One could argue that the title overstates what is achieved in the paper. I wholeheartedly agree that this study is a significant step forwards but I don't think that at this stage it actually 'improves' climate simulations. The model is still idealized and does not compare to observations and other realistic climate models. I understand that titles like these are common at journals like NatComms but I would prefer a more accurate title that highlights the achievements of the paper, namely stable simulations using coarse-grained high-res data.

In response to the reviewer comment we changed the title of the manuscript to: 'Use of a subgrid parametrization learned from a high-resolution atmospheric model for simulation of climate'.

Final discussion: I think that the discussion could be enhanced. In particular, there is no mention about the RF vs NN question. I think that a sentence or two highlighting the advantage of RFs over NNs is certainly warranted since, to my knowledge (having last

talked to Noah and Chris a couple of months ago), NNs are still struggling with stable simulations. Plus, there seems to be no real reason for using NNs, since their advantages (differentiability, complex architectures) have not been used for parameterizations. Another missing discussion, in my opinion, is the difference of your coarse-graining method with the one of Brenowitz and Bretherton. I think this would be of great interest to many readers. This could go at the end or when describing the coarse-graining, which brings us to the next point. For example, one advantage of your method is that it does not require you to do any temporal or horizontal differencing.

In response to this comment, we added text to the discussion section discussing the possible advantages of RF over NN (lines 290-297) and we also highlights the main differences in how we calculated and used the subgrid terms compared to previous studies (lines 281-287).

Main text description of coarse-graining and RFs (line 90ff): A lot of the details about the method are deferred to the supplement. I believe that it would make sense to add more detail to the main text, as currently it is very high-level. In particular, I think the text could benefit from a more in-depth coverage of: a) Why are two RFs necessary? This is actually not entirely clear to me even with the Supplement. You mention that RF2 has different inputs (upper-levels cut off) to save time but would it really be computationally impossible to have a single RF? b) What are the inputs/outputs? I think moving the input and output vectors from the supplement (e.g. supp. line 132) to the main text would make the text easier to understand. c) What is diffusivity? Where does what you predict show up in Eqs. S1-S3? This point could be covered in the Supplement only.

In response to this comment, we added text to the main paper to describe the inputs and outputs of the two RFs (lines 122-156). The reason we use two RFs is because different physical processes are a function of different fields (e.g., the surface fluxes depend on the horizontal wind variables, while microphysics or vertical advection is independent of the horizontal wind variables) and the processes included in RF-diff are mainly active in the lower troposphere, and this reasoning is now explained on lines 122-125.

To clarify what we mean by the turbulent diffusivity we added text in the Supplementary Information (lines 143-145).

Using $|y|$ as input: The distance to the equator was used as a proxy for some latitude-dependent variables. I wonder, however, if there is a slight potential for 'cheating'. The data has a very strong latitudinal structure, so the RF could simply learn that structure rather than basing its predictions upon the actual SST, etc. I wonder what would happen if SST, albedo and insolation were directly fed into the RF. Most likely the differences would be small but maybe it would be good to at least add a sentence discussing this if the actual experiments are too difficult to run. For real geography cases, $|y|$ is not sufficient anyway.

We agree with the reviewer that in case we want to use the same algorithm in a different and less symmetric configuration we would train the RF parameterization using the relevant variables (SST, surface albedo and solar insolation) as features and not the distance from equator. In the context of the current work, since RF is a decision tree based algorithm, where in each split of the trees a cut-point is chosen to divide the data to two different sub-

data sets, using the distance from the equator or the SST, surface albedo and solar insolation are equivalent. In response to this comment, we added text to the main paper explaining these points (lines 141-145).

To demonstrate that the RF is not using the distance-from-equator feature as a proxy for different circulation regimes, we have re-trained the RF parameterization without using the distance from equator as a feature and found that results are almost identical (see Table S1 for the offline results of this test). This is reassuring since much of the subgrid physics (vertical advection, cloud and precipitation microphysics, longwave cooling) does not depend on or relatively weakly depends on the variables that $|y|$ is a proxy for (in the case of longwave cooling, the RF can use temperature at the lowest model level as a proxy for SST). Furthermore, we plot here the mean precipitation from a simulation with this alternative RF parameterization (Fig. 1a). We have also added some results for generalization across latitudes in the new section on robustness of the RF parameterization.

Small comments and typos:

Line 35: 'A NN' instead of 'An NN'

An NN is the right form since NN is an initialism and the pronunciation of the first letter 'N' starts with a vowel sound.

Line 62: 'ocean' Maybe it would be clearer to explicitly mention the word 'aquaplanet' which has been commonly used.

We now say it is in an aquaplanet configuration (line 83).

Line 106: 'necessary to include multiple time steps' This also would not be possible using a RF since it's not differentiable, right? I also found the following sentence a little unclear: 'different approach', 'different machine learning algorithm'. I think it would be clearer if you explicitly names what these different approaches are (see my comment on mentioning the differences in coarse-graining above).

We agree with the reviewer that including multiple time steps would require a different machine learning approach. Following the comment of the reviewer, and after revising the paper, we have decided to omit this sentence. However, we do highlight differences in our approach compared to previous work in the discussion section (lines 281-297) as suggested by the reviewer.

Line 153: At the beginning of this section you ask the question whether there is a scale particularly suitable for parameterization. So what is your conclusion based on your results? Is it $8x = 100km$? Maybe add a sentence or two of discussion.

In the revised Figure 3, the online accuracy of the parameterization increases monotonically with decreasing grid spacing (x4-RF has the highest accuracy of the coarse-graining factors we explored). Therefore, our conclusion in this section is that "there is a clear overall improvement for the RF parameterization in performance as grid spacing decreases" (lines 232-234).

Reference 17. My paper now has a slightly different, hopefully less confusing, title: 'Coupled online learning as a way to tackle instabilities and biases in neural network parameterizations'

Reference was updated.

Supplement

Line 33: What height is the model top?

The height at the top level of the model is 28695m, and this information was added to the Supplementary Information at line 36

Line 86: Just out of curiosity, how did you technically compute the resolved tendencies? Did you coarse grain your high-res fields and then feed them to low-res SAM to compute the tendencies?

Yes, we first calculated the coarse-grained fields and then calculated the subgrid tendencies. We did this in matlab but it would be better to implement this as an offline program in fortran90 in future work. We added to the Supplementary Information some more details about our approach for this (lines 103-112).

Line 103: 'not predicted above'. I have several questions about the cut-off of radiative predictions. First, does the switch from RF to SAM lead to any noticeable discontinuities? Second, why does including the RF heating rates lead to a drift? It would be great if you could dig a little deeper into this topic and maybe discover some mechanism. In our PNAS paper we also had strong biases in the stratosphere. I was never able to find out exactly why this was happening. But if you are having similar issues, maybe there is a more systematic cause?

In order to check if the diffusivity cutoff leads to discontinuities, we calculated the absolute value of the diffusivity difference between the two levels where the cutoff occurs (in the x8-RF simulation), and we compared this difference to the diffusivity difference between these two levels if the diffusivity would be calculated using SAM code only (in the x8-RF simulation). We find that the difference in the RF case is actually smaller ($3.8\text{m}^2/\text{s}$) compared to the case of using the diffusivity calculated in SAM ($6.4\text{m}^2/\text{s}$). This may be because the RF case underestimates the variance in the diffusivity. In any case the cutoff does not seem not to lead to any discontinuities.

We are not completely sure why including the RF heating rates in the stratosphere leads to a drift in the stratospheric temperature, but it may be because of insufficient coupling between the stratosphere and the troposphere and the use of a column-based parameterization. We now mention this possibility in the Supplementary Information (lines 130-133) and we also refer to Rasp et al PNAS (lines 128-129).

Line 199: It would be very interesting if you stated somewhere how you implemented the RF in SAM. Did you hardcode the RF in Fortran or did you interface with Python?

The training process was done in python, while a Fortran module was used to read the RF into SAM and provide online predictions. This is now clarified in the Supplementary Information (lines 227-229). The Fortran code is available in our osf archive and was also made available in the zenodo archive for O’Gorman and Dwyer, JAMES, 2018.

Line 249: Energy conservation error: It might be helpful to state somewhere the mean absolute magnitude of the vertically integrated tendency as a reference for the residual values. Without context it is hard to appreciate how accurate the RF is.

In response to this comment we added the standard deviation of the vertical integral of the true energy tendencies (which is 64.76W m^{-2}) to the Supplementary Information (lines

291-293)

Line 269: 'timestep' Did you try running this parameterization at a coarser time step?

We have only made modest changes to the time step because the turbulence scheme in SAM uses explicit timestepping and this limits the timestep through the CFL condition as now mentioned on lines 313-315 of the Supporting Information. Global climate models use different numerical approaches for diffusion (e.g., fully backwards implicit time stepping) to avoid this problem. SAM could be modified to deal with this issue in future work.

Line 292: 'Third, ...' I personally would like a more in-depth discussion of the instabilities. As far as I understand Noah's instabilities can be traced back to a wrongly reversed causality, where upper level Q perturbations lead to strong tropospheric heating. So you suspect the same mechanism? Why does adding q_p prevent this?

We do not know what causes the instability for this alternative form of the parameterization in which q_p is not included and when the stratospheric tendencies are included. Brenowitz and Bertherton omitted inputs from the NN, while we do not omit inputs, but instead we do not correct the vertical advection and sedimentation in the stratosphere. We now say (lines 179-181) that this alternative parameterization "requires certain outputs to be set to zero above to avoid a deleterious feedback possibly related to an issue of causality when q_p is not evolved forward in time". This parameterization is effectively making q_p diagnostic ($dq_p/dt = 0$) and this may lead to issues of causality. The parameterization without using q_p in SAM is not the main focus of this study, and we will leave the investigation of these question for future work.

Table S2: Also show lat-z mean plots of R2.

Such a figure is shown in the SI (figure S2)

Figure 1: Zonal- and time-mean precipitation and for (a) the x8-RF standard simulation (red) and the x8-RF simulation without the distance from equator as a feature (dotted black), and (b) the x8-RF standard simulation (red) and the x8RF without using the mapping v to $-v$ in the southern hemisphere (dotted black)

References

Response to reviewer 3

The authors use 3D high resolution model output from SAM to train machine learning parameterisations using the Random Forests technique. In implementing their machine learned parameterizations in online testing, the authors are able to create realistic and stable simulations of the climate. These results are novel, interesting and worthy of publication in Nature Communications. The manuscript is very well written and in a style consistent with the journal. The figures are also very well presented (especially Fig 1.). I would be happy to read the manuscript a second time should it be needed. A relevant caveat for my review of this manuscript is to state that I have not implemented any ML techniques. As such, I can not comment on the implementation choices or the subtleties of the method. However, I am experienced with modelling convection and model development more generally and it is this experience on which my review is based. I have listed a few minor comments and a longer list of clarification items. In the majority of cases, my comments are from the perspective of helping non-ML experts read the manuscript and interpret the significance of the results. I have also included a list of editorial comments. There is no need to respond to any of the editorial comments in your response to reviewers. The majority of these editorial comments are suggests and can be changed at the authors discretion.

We thank the reviewer for their careful reading and the useful comments. Please see our reply to all the comments below. As a result of the reviewer comments we have made several modifications in the manuscript (described below).

Minor Comments

1. Abstract - I think you should mention these are online parameterisations right at the start. This is an important piece of info.

We mention that in the abstract that “The parameterization leads to stable simulations at coarse resolution that replicate the climate of the high-resolution simulation.” This implies an online parameterization. Ideally we would have introduced the idea of online versus offline in the abstract. But since the abstract word limit is 150 words and due to other requests by reviewers it was difficult to accommodate more information into the abstract.

2. The manuscript would benefit from a sentence or two explaining the RF technique (or at a minimum pointing the reader to some refs), citing the code used if possible, and a longer introduction to the RF techniques in the supplementary.

We have added a sentence (lines 62-64) to the introduction to say that “An RF is an ensemble of decision trees, and the predictions of the RF are an average of the predictions of the decision trees (Breiman, 2001; Hastie et al., 2001)” which introduces the concept and provides two standard citations. Furthermore, in the Supplementary Information we refer to the exact code version of the RF we use, namely the RF is taken from the RandomForestRegressor class from scikit-learn package (Pedregosa et al., 2011) version 0.21.2. Since random forest are widely-used algorithm, we think that it is suitable to refer to the two cited references in the main text for further details.

3. L155-162 I think the discussion could be longer. I feel you undersell the significance and importance of your simulations.

In response to this and other comments, we have substantially extended the discussion section.

4. Sup L34-35: You mention that the time step is halved when the model is unstable. Is the model often unstable? At what resolutions was this a problem? Was it only early in the simulations or often crash later on?

Halving the time step is an adaptive time stepping technique used in the original SAM version. We don't mean to imply the simulations become unstable. All the simulations we ran (in all resolutions) are always stable and never crash. We have modified this sentence to say "and this is adaptively reduced as necessary to prevent violations of the CFL condition" to avoid confusion (lines 36-37).

5. Sup L39-42 The hypohydrostatic rescaling is usually applied to reduce the resource requirement to run these simulations. Given the ML parameterisation you develop is much faster than traditional parameterizations, why use this rescaling? Was this needed to make more (or longer) training data sets?

Using hypohydrostatic rescaling greatly reduces the computational expense of the original hi-res simulation and the amount of disk storage needed to store output (roughly 4TB with hypohydrostatic rescaling versus 64TB without). Importantly, using hypohydrostatic rescaling reduced the amount of resources needed to perform calculations based on the hi-res output files. For example, calculating the resolved and subgrid tendencies consumes a lot of computation time, and the hypohydrostatic rescaling reduced the computation time significantly.

Clarifying Comments

abstract

1. L8: I think you should mention the model domain or that it is idealised to avoid confusion with 3D global high-res GCMs (will the full complexity of such simulations).

In response to this comment we now mentioned that the model is idealized in the abstract (line 8). Since the abstract is limited to 150 words it is difficult to make a more detailed statement.

Introduction

1. Is 'learn' and 'train' the same thing? They are used interchangeably in places (eg L 5, 7, 12, 26, 56). In my mind we use training data to train a ML scheme and that 'learning' is the resulting statistical model that is implemented as the scheme.

We agree with the reviewer that we had loosely used the two terms interchangeability. In response to this comment we reviewed and modified the text such that we use 'train' in the context of the training process and 'learn' in the context of the resulting model.

2. I think you should state early more clearly how this study is similar and different to O'Gorman and Dwyer (2018). Or at a minimum state that the method is the same but the implementation is different. This will help make the new results of your study clearer.

The main similarity between this study to the work of O’Gorman and Dwyer is the fact that both studies use the same ML algorithm for parameterization of subgrid atmospheric processes. The paper had already said on lines 68-70 that “When an RF was used to emulate a conventional convective parameterization, it was found to lead to stable and accurate simulations of important climate statistics in tests with an idealized GCM (O’Gorman and Dwyer, 2018).” In response to the reviewer comment, we added a sentence on the following line to clarify that “Thus RFs are promising for use in learning parameterizations of atmospheric processes, but they have not yet been used to learn subgrid processes from a high-resolution atmospheric model.” (lines 70-72)

3. L28-31 I was under the impression that ML in warmer climates fail to simulate realistic climates (when trained on historical data). Do you mean that training should include warmer climates and then it is trivial? Would you need to include a very large temperate range in your training data?

Yes, you are correct to say that ML parameterizations trained only on historical (colder) data set failed to generalize to warmer climates (O’Gorman and Dwyer, 2018; Rasp et al., 2018). Therefore, we state in the paper (lines 33-35) that “Training on both the control climate and a warm climate is needed to simulate a warming climate using an ML parameterization”. The temperature range that is needed in the training data to be able to generalize depends on the future climate. A rule of thumb is that it is necessary to have training samples that have similar temperature distribution as in the future climate.

Results

1. Did you train the data on the same system that SAM was run on? Or was the training done on GPUs?

The training process was done on the same system and a GPU is not needed as the training process is relatively fast for an RF. It typically takes less than 1 hour if done on 10 cores simultaneously which we now mention on lines 218 of the Supporting Information.

2. The double ITCZ in high-res simulations vs single ITCZ in the x8 runs is very interesting. Has this been seen in SAM before?

Bretherton and Khairoutdinov (2015) analyzed quasi-global simulations with SAM and found a single ITCZ at both high resolution and coarse resolution, whereas we find a single ITCZ at coarse resolution and a double ITCZ at high resolution. However, they used a different domain (wider in the zonal direction but narrower in the meridional direction), the full Coriolis parameter rather than an equatorial beta plane, and did not use hypohydrostatic rescaling. It is unclear which aspect (or other minor details) caused the difference in behavior, but the setup used (prescribed SST at equinox) can easily switch from single to double ITCZ as mentioned on lines 94-102 of the paper.

3. L126-129 I am confused about the use of the alternative RF parameterization and the different treatment of q_p . Is this only for speed or are there other advantages? If this has similar results and is cheaper, then why not is it instead?

There are several potential advantages of using the second formulation described in the Supplementary Information. As the reviewer mentions, this formulation has the potential of being much faster. Furthermore, many climate models do not use q_p as a (prognostic)

variable, and therefore using an ML parameterization in these models requires a parameterization that does not use q_p as an input. This is now clarified in the Supplementary Information on lines 311-315. The reason we chose to describe the parameterization that uses q_p in the main text is that it is considerably more straightforward to explain, doesn't require excluding certain tendencies for stability, and is slightly more accurate in terms of its climate.

4. Fig 3: I would reorder the subplots so that the second row is the qT tendency contours (the subplots c and e would also benefit from units or color bar table or subheading) and the third row is P. If this is not possible, then I suggest moving d and f to a new plot so that R2 and zonal precip are separated. If you are aiming for a three fig paper, then perhaps put the Precip plots in fig 2?

The reason we chose to use the current order of the panels is that the left panels all describe offline performance while the right panels all describe online performance. We have restructured the caption to make this easier to follow. Due to the requests of the other reviewers we added 2 additional plots to the manuscript and we decided not to split this figure to two different figures.

Supplementary

1. In equ S1 and S2 there are large-scale prescribed terms. Could you add a comment in the appendix about why these are not in your formulation? Could you also add a comment on why you included the sublimation term (S) in equation S2.

In our simulations the large scale terms are zero and therefore they are not included in equations S1 and S2. The reason we included sedimentation in equation S2 is because unlike the other microphysical processes (such as autoconversion or evaporation), which affect q_p and q_T , the sedimentation also affects the energy (h_L) and is used in the diagnosis of surface precipitation (see equation S7). In response to this comment we added text explaining that we do not prescribe large-scale tendencies to the simulations we run (lines 31).

2. Sup L26-29 Could you explain what the partition function (ω) is? I assume this means how the water species are separated but I am not sure.

Yes, ω is the function determining how water is separated into liquid and ice phases. In response to this comment we clarified this in the text (lines 27-28 for precipitating water and line 30 non-precipitating condensate).

3. Sup L34. Is 24 sec the default for SAM or the default time step for your study?

24 seconds was the time step used in the hi-res simulation, and this value was not changed in any of the simulations we describe

4. Sup L47-49. What is the significance of saving snapshots every 3 hours? Is hourly to data heavy and 6 hourly not helpful? I don't see the significance of 3 hours and I assume there is no sensitivity to this frequency given it is a snapshot (given the simulations are stable).

There is no special significance of saving snapshots every 3 hours. Using a very short time interval between snapshots might increase the correlations between training samples (which we would like to avoid), but require a shorter hi-res run to create the same amount of training data. On the other hand, increasing this time interval would reduce the correlations between

snapshots but would require a longer simulation in order to produce the same amount of training data. The 3-hour interval is a good compromise.

5. Sup L127-127: could the local time/space assumption be relaxed so that the system could have memory or organisation?

Yes, it is possible to introduce memory/longer range interaction, and using such features might lead to an improvement in the performances of ML parameterization. We intend to further investigate in the future if including memory or non-local inputs improve the performance of the parameterization and this is now mentioned as a possible avenue for future research at the end of the discussion section (lines 302-304).

6. Sup L 160: Why is the variability important but not the mean? What then sets the mean?

Since the RF uses the mean square error as its evaluation criterion (splitting criterion), using outputs that have very different scales would imply that the RF tries to predict correctly some outputs more than other outputs. The rescaling procedure for the outputs described in L160 aims to make sure the different type of outputs (e.g., h_L , q_T and q_p in RF-tend) are weighted evenly (which is a choice, but other possible choices might make sense as well). Rescaling the outputs to have zero mean and variance of one is a common choice, but in the context of RF not subtracting the mean outputs would have little effect on the training procedure. We stress that after training the RF the outputs are inverse scaled back to their original values.

7. Sup L 170-172: Might be worth mentioning in here that low correlations between samples is the goal - to non ML people low correlations would read as a bad thing.

We now say “To make the samples more independent, ...” on lines 195-196

8. Sup L 178: Could you explain what hyperparameters are used before describing their tuning?

We now say “ Different hyperparameters governing the learning process and complexity of the RFs may be tuned to improve performance.” on lines 203-205

9. Sup L 182: Could you add that 'R' is the correlation to help people interpret R2 (might be good to say this in the manuscript as well)?

The coefficient of determination is calculated as $R^2 = 1 - \frac{\sum_i (y_i - \bar{f}_i)^2}{\sum_i (y_i - \bar{y})^2}$, where y_i is the observed data sample i , f_i is the predicted data and an overbar is the mean over samples. The coefficient of determination can be negative for a poorly fit statistical model, and it is not generally equal to the correlation coefficient squared except for special cases such as ordinary least-squares regression. Thus we do not say that R is the correlation to avoid confusion.

10. Sup L269: what do you mean by a 'fast variable'?

In response to this comment we changed the term in the Supplementary Information to 'variable that changes on short time scales' on line 310

11. Equ S11: Could you check the sign of the last term please? The P_{tot} looks fine but should the other two terms have opposite sign?

Did you mean equation S1 ? (In equation S11 P_{tot} does not appear). If you mean equation S1, the term $(\frac{\partial h_L}{\partial t})_{rad}$ refers to the radiative heating tendency (this is the sign convention we

use). The other term (in equation S1) is $-\frac{1}{\rho_0} \frac{\partial}{\partial z} (L_n S)$ and we think that as it appears is the correct sign (convergence of sedimentation flux should decrease h_L since h_L decreases with increases in condensate).

References

- Breiman, L., 2001: Random forests. *Machine learning*, **45** (1), 5–32.
- Bretherton, C. S., and M. F. Khairoutdinov, 2015: Convective self-aggregation feedbacks in near-global cloud-resolving simulations of an aquaplanet. *J. Adv. Model. Earth Sys.*, **7** (4), 1765–1787.
- Hastie, T., R. Tibshirani, and J. Friedman, 2001: *The elements of statistical learning*. 2nd ed., Springer, 745 pp.
- O’Gorman, P. A., and J. G. Dwyer, 2018: Using machine learning to parameterize moist convection: Potential for modeling of climate, climate change, and extreme events. *J. Adv. Model. Earth Sys.*, **10** (10), 2548–2563.
- Pedregosa, F., and Coauthors, 2011: Scikit-learn: Machine learning in python. *J. Mach. Learn. Res.*, **12** (Oct), 2825–2830.
- Rasp, S., M. S. Pritchard, and P. Gentine, 2018: Deep learning to represent subgrid processes in climate models. *Proc. Natl. Acad. Sci. U.S.A.*, **115**, 9684–9689.

REVIEWERS' COMMENTS:

Reviewer #1 (Remarks to the Author):

I am more than satisfied by the authors' admirable responses and improvements to the original manuscript including its broader message & linkages to other literature in this space. I think this draft would be great to publish in NCC.

Thanks for answering all of my questions! The updates to the scale sensitivity results and emphasis on the RMSE over R^2 in understanding those results are really nice. I also think I finally understand how you did this. It is a huge accomplishment.

Thanks again,

-- Mike Pritchard.

Reviewer #2 (Remarks to the Author):

The authors thoroughly addressed all of my concerns and questions. I am happy for the paper to be published as is.

Best,
Stephan Rasp

Reviewer #3 (Remarks to the Author):

Please see the attached pdf for my review. Penelope Maher.

Review of ‘Use of a subgrid parametrization learned from a high-resolution atmospheric model for simulation of climate’

by Janni Yuval and Paul O’Gorman

Nature Communications manuscript number NCOMMS-19-42290-T

Revision Number: 2

Recommendation: Accept

Reviewer Name: Penelope Maher

1 Summary of the Review

The authors have prepared a well considered revised manuscript with significant changes to the text and figures. The authors have also sufficiently address my comments in the first revision. I recommend the manuscript be accepted for publication. I have only a few editorial comments for the authors to consider. These could easily be addressed during copy editing and should not hold up the review process.

2 Editorial comments

The following editorial comments are suggestions only. I am happy for the authors to execute their own discretion on all of these. There is no need to respond to any of these points in your response to reviewers.

1. L41-43: suggest rewriting this sentence as it is a bit disjointed.
2. L51 or near by: it would be helpful to mention what coarse-graining is.
3. L138 and else ware: suggest replace ‘(text S1)’ with ‘(see section S1)’ or similar.
4. L281: suggest rewriting ‘is different in important’ to ‘is unique in three key ways’ or similar.
5. Equ S12: in my first review, I suggested checking the sign of a term in equ S11 - this was an error in labeling and should have been S12. At present I do not have access to my notes where I worked through this, so I can’t clarify any further on my comment. If the authors are happy with the formulation, that is sufficient for me.

Response to reviewers

1 Reviewer 1

I am more than satisfied by the authors' admirable responses and improvements to the original manuscript including its broader message and linkages to other literature in this space. I think this draft would be great to publish in NCC. Thanks for answering all of my questions! The updates to the scale sensitivity results and emphasis on the RMSE over R^2 in understanding those results are really nice. I also think I finally understand how you did this. It is a huge accomplishment. Thanks again, – Mike Pritchard.

We thank the reviewer for their very useful comments throughout the revision process.

2 Reviewer 2

The authors thoroughly addressed all of my concerns and questions. I am happy for the paper to be published as is. Best, Stephan Rasp

We thank the reviewer for their very useful comments throughout the revision process.

3 Reviewer 3

The authors have prepared a well considered revised manuscript with significant changes to the text and figures. The authors have also sufficiently address my comments in the first revision. I recommend the manuscript be accepted for publication. I have only a few editorial comments for the authors to consider. These could easily be addressed during copy editing and should not hold up the review process.

We thank the reviewer for their very useful comments throughout the revision process. Please see our a point-by-point reply comments made by the reviewer.

Editorial comments

The following editorial comments are suggestions only. I am happy for the authors to execute their own discretion on all of these. There is no need to respond to any of these points in your response to reviewers.

1. L41-43: suggest rewriting this sentence as it is a bit disjointed.

We agree with the reviewer that the sentence was unclear. Following this comment we rewrote the sentence as follows: “Without such modifications, it may be better to turn off some conventional parameterizations of deep convection for a range of grid spacings that are too close to the convective scale.”

2. L51 or near by: it would be helpful to mention what coarse-graining is.

Following this comment we added in parenthesis that coarse-graining refers to spatial-averaging

3. L138 and else ware: suggest replace '(text S1)' with '(see section S1)' or similar.

Following this comment and comments made by the editor we replaced all reference to the supplementary information, figures and tables to fit nature style.

4. L281: suggest rewriting 'is different in important' to 'is unique in three key ways' or similar.

We decided to keep the current phrasing.

5. Equ S12: in my first review, I suggested checking the sign of a term in equ S11 - this was an error in labeling and should have been S12. At present I do not have access to my notes where I worked through this, so I can't clarify any further on my comment. If the authors are happy with the formulation, that is sufficient for me.

We rechecked the signs in the equation and we think that how it is written is the correct form.